# Revealing Scaling Paradox in Large-scale Time Series Models: Implications for More Efficient and Accurate Forecasting

Xin Qiu [* 1 2]    Junlong Tong [* 1]    Yirong Sun [* 1]    Yunpu Ma [3]    Xiaoyu Shen [1]

## Abstract

Large-scale models are at the forefront of time series (TS) forecasting, dominated by two paradigms: fine-tuning text-based Large Language Models for TS (LLM4TS) and training Time Series Foundation Models (TSFMs) from scratch. Both approaches share a foundational assumption that scaling up model capacity and data volume leads to improved performance. However, we observe a ***scaling paradox*** in TS models, revealing a puzzling phenomenon that larger models do *NOT* always achieve better performance. Through extensive experiments on two model families across four scales (100M to 1.7B parameters) and diverse data (up to 6B observations), we rigorously confirm that the scaling paradox is a pervasive issue. We then diagnose its root cause by analyzing internal representations, identifying a phenomenon we call *few-layer dominance*: only a small subset of layers are functionally important, while the majority are redundant, under-utilized, and can even distract training. Based on this discovery, we propose a practical method to automatically identify and retain only these dominant layers. In our models, retaining only 21% of the parameters achieves up to a 12% accuracy improvement and a $2.7\times$ inference speedup. We validate the universality of our method on 8 prominent SOTA models (LLM4TS and TSFMs, 90M to 6B), showing that retaining less than 30% layers achieves superior accuracy in over 95% tasks. We release our code at LLM4TSF-Scaling.

## 1. Introduction

Time series (TS) forecasting is a critical task in real-world applications, ranging from financial analysis to climate monitoring (Lim & Zohren, 2021; Tong et al., 2023b). However, modeling long-term temporal dependencies and diverse patterns across domains remains a fundamental challenge. Inspired by the success of large-scale models, the TS community has explored two major paradigms. The first, Large Language Models for Time Series (LLM4TS), leverages the existing capabilities of text-based pretrained LLMs by aligning TS embeddings into the text modality, whether at the word (Jin et al., 2024; Pan et al., 2024), sentence (Liu et al., 2024a; 2025a), or latent-space levels (Liu et al., 2025b; Hu et al., 2025). The second, Time Series Foundation Models (TSFMs), pre-trains models from scratch purely on massive TS data to learn universal representations (Woo et al., 2024; Liu et al., 2024c). Despite their methodological differences, both approaches share the underlying assumption that scaling up model capacity and data volume will improve performance (Zhang et al., 2024; Liu et al., 2025c).

However, this foundational assumption is challenged by a series of scattered observations. For instance, employing full-scale LLMs as backbones does not necessarily yield better performance than truncating lower layers (Zhou et al., 2023; Liu et al., 2025a; Hu et al., 2025; Qiu et al., 2026), and advanced LLMs often fail to show expected advantages over weaker ones (Jin et al., 2024). Likewise, in TSFMs, larger models such as Moirai$_{Large}$ often lag behind smaller ones like Moirai$_{Small}$ (Goswami et al., 2024; Das et al., 2024; Liu et al., 2025c; Qiu et al., 2026), with performance gains being marginal at best.

These findings, however, have so far been treated as isolated or scattered observations. They have lacked a in-depth investigation to determine if they are a systemic issue. In this paper, we are the first to formalize this problem and conduct a comprehensive study. We term these collective observations the *scaling paradox*, a phenomenon that stands in sharp contrast to the scaling laws observed in other domains, as evidenced by the results in Tab. 1.[1]

---

[*]Equal contribution [1]Ningbo Institute of Digital Twin, Eastern Institute of Technology, Ningbo, Zhejiang 315200, P. R. China [2]Zhejiang University [3]LMU Munich. Correspondence to: Xiaoyu Shen <xyshen@eitech.edu.cn>.

*Proceedings of the $43^{rd}$ International Conference on Machine Learning*, Seoul, South Korea. PMLR 306, 2026. Copyright 2026 by the author(s).

[1]The results are reproduced from (Pan et al., 2024; Liu et al., 2025c).

*Table 1.* Scaling paradox (Scaling model leads to higher or negligible reductions in Mean Absolute Error)

| | Sundial$_{\text{Small}} \to$ Sundial$_{\text{Large}}$ | Moirai$_{\text{Small}} \to$ Moirai$_{\text{Large}}$ | Time-LLM$_{\text{GPT-2}} \to$ Time-LLM$_{\text{LLaMa}}$ |
|---|---|---|---|
| ETTh1 | $+0.002$ 📶 | $+0.020$ 📶 | $+0.016$ 📶 |
| ETTh2 | $0_{\downarrow 0\%}$ | $+0.006$ 📶 | $+0.002$ 📶 |
| ETTm1 | $-0.019_{\downarrow 5\%}$ | $+0.006$ 📶 | $+0.013$ 📶 |
| ETTm2 | $-0.009_{\downarrow 3\%}$ | $+0.006$ 📶 | $-0.007_{\downarrow 2\%}$ |
| ECL | $-0.009_{\downarrow 1\%}$ | $-0.004_{\downarrow 1\%}$ | $+0.009$ 📶 |
| Weather | $+0.004$ 📶 | $-0.008_{\downarrow 3\%}$ | $-0.003_{\downarrow 1\%}$ |

To explore this paradox, we compare two architectural model families, each covering four scales ((i)Tiny, (ii)Small, (iii)Base, and (iv)Large) with parameter sizes ranging from under 100M to over 1.7B. These models are trained on datasets spanning data volumes from 60K to 6B observations and up to 41 data distributions, following both single-dataset and cross-dataset learning strategy (Chang et al., 2025). We evaluate the in-domain and out-of-domain forecasting abilities of the 8 trained models across 16 data distributions. The results confirm that the ***scaling paradox*** is a pervasive phenomenon across different models and data distributions, and it is particularly pronounced in LLM4TS. After confirming its existence, we conducted a careful analysis of the models' internal representations to diagnose the root cause. Further analysis reveals the paradox shows weak correlations with data volume, diversity, and learning strategy. Instead, we find the cause is *"few-layer dominance"*: (1) *not all layers contribute effectively to the prediction process*, and (2) *retaining only a few true executors can achieve performance comparable to the full model*. This suggests that most layers are redundant and the parameters are largely under-utilized. In contrast, we find they can even distract the training process, leading to performance degradation.

Based on this discovery, we propose a practical method to identify and retain only critical layers. Specifically, utilizing only **21%** of the parameters achieves superior performance, leading to a **2.7×** faster inference speed and up to a **12%** improvement in accuracy, while the performance degradation in the worst case is as small as **0.6%** in our model families. Moreover, we validate the universality of our method by applying it to **8 famous TS models** (4 LLM4TS & 4 TSFMs). The model architectures cover three paradigms: encoder-only, decoder-only, and encoder–decoder, while the TS embeddings operate at three levels: point-wise, patch-wise, and variable-wise. Their model sizes range from 90M to 6B. The TSFMs vary in their pre-training scales from 100B to 1,000B observations, and LLM4TS differ in prompt designs and alignment strategies (word level, sentence level, and latent-space level). Experiments are conducted on 13 data distributions with three input lengths and four forecasting horizons. Remarkably, by retaining less than **30%** of the critical layers on average, our method achieves comparable or even superior forecasting accuracy in over **95%** of tasks, with the average accuracy loss in the remaining cases kept

below **0.5%**. Our main contributions are threefold:

1. We provide the first analysis of a *scaling paradox* in large-scale TS models, rigorously demonstrating that *"larger is not always better"*.

2. We diagnose the root cause of this paradox, identifying *few-layer dominance*, where only a small subset of layers are functionally important.

3. We propose a practical *pruning method* that identifies and retains the critical layers, producing smaller and faster models that even improve forecasting accuracy.

## 2. Background

In this section, we provide the necessary background, including general architectures and core components of LLM4TS and TSFMs. Based on them, we design the model families employed in the experimental analysis of our work.

**Architecture of LLM4TS.** The main challenge of LLM4TS is to represent TS in a way that LLMs can effectively process, while minimizing the modality gap through proper strategies. Its **input embed** module consists of a TS embed module, which maps TS into embedding, and a tokenizer that injects prompts into the LLM. The **LLM4TS backbone** then aligns the TS embeddings with the LLM representation space through fine-tuning. The aligned TS embeddings are passed through a **prediction head** to generate the prediction results, with the model optimized by minimizing the MSE objective.

**(I)Input embed.** Patch-wise (Nie et al., 2023) and variable-wise (Liu et al., 2024b) TS embed are widely used to transform TS into embeddings. The LLM tokenizer encodes the prompts and injects them into the model, allowing the LLM to exploit its intrinsic understand and reasoning advantages. **(II)LLM4TS backbone.** The backbone can be instantiated with LLMs such as GPT-2 (Radford et al., 2018), LLaMA (Touvron et al., 2023), or Qwen-3 (Yang et al., 2025). By fine-tuning, the model performs alignment at the word, sentence, or latent-space levels, strengthening its understanding of historical patterns and reasoning over future steps (Pan et al., 2024; Liu et al., 2025a; Hu et al., 2025). **(III)Prediction head.** It serves as a decoder that transforms TS embeddings into predictions. It typically adopts an autoregressive or a parallel decoding strategy. The parallel decoding paradigm mitigates error accumulation and lowers latency (Zhou et al., 2021; Zeng et al., 2023).

**Architecture of TSFMs.** TSFMs feature a more flexible architecture, consisting of a **TS tokenization** module and a **TSFM backbone**. Instead of adapting pre-trained LLMs, they design specialized structures and temporal modeling modules to capture dependencies offering greater adaptability to diverse forecasting tasks (Chen et al., 2023). TSFMs

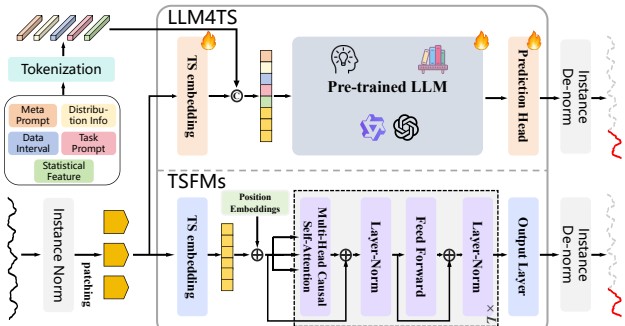

*Figure 1.* Overall structure of our LLM4TS and TSFMs families. Each family consists of four models at different scales. LLM4TS fine-tunes only a small subset of parameters to transfer the extensive knowledge of LLMs to TS tasks, while incorporating prompts to enhance learning. TSFMs adopt a decoder-only architecture that predicts the next token to forecast future.

can also employ more diverse supervision losses, allowing greater flexibility in optimizing objectives. **(I)TS tokenization.** Besides patch-wise, TSFMs may encode TS data through scaling or quantization, adopt point-wise tokenization to preserve the completeness of temporal information (Ansari et al., 2024; Liu et al., 2025c). **(II)TSFMs backbone.** Beyond LLM4TS adopts a decoder-only architecture to predict the next token, some TSFMs follow an encoder–decoder paradigm to update temporal tokens. Meanwhile, an explicit prediction head is unnecessary, since the backbone itself can map tokens back to numerical values through autoregressive sampling or multi-scale forecasting (Das et al., 2024; Shi et al., 2025).

## 3. Scaling Paradox

In this section, we investigate the *scaling paradox*. Sec. 3.1 provides an overview of our work, and the following sections investigate *five potential factors* affecting forecasting performance, and analyze external factors.

### 3.1. Analysis Framework

We systematically investigate key factors of TS forecasting task, including architecture, model size, data volume, distribution, and learning strategies, to illustrate how they influence forecasting and the emergence of scaling paradox.

**Model Architecture.** We employed two representative architectural designs that are prevalent in contemporary TS research (Zhou et al., 2023; Das et al., 2024), which also avoids confounding effects from introducing too many variables at once. To mitigate semantic shifts from uneven data distributions, we apply instance normalization and patch-based embedding (Nie et al., 2023). Since the variables are mostly independent (Chen et al., 2025), we adopt a channel-independent design to reduce overfitting. For the LLM4TS

*Table 2.* Main model parameters. For LLM4TS, the learnable parameters exclude the frozen components within the underlying LLMs, while for TSFMs, since no modules are frozen. "–" indicates no pre-training (trained from scratch).

| Scale | Model family | Layers | Backbone | Channels | Token Level |
|---|---|---|---|---|---|
| Tiny | LLM4TS / TSFMs | 6 / 6 | GPT-2 / - | 768 / 768 | Patch / Patch |
| Small | LLM4TS / TSFMs | 12 / 12 | GPT-2 / - | 768 / 768 | Patch / Patch |
| Base | LLM4TS / TSFMs | 28 / 28 | Qwen-3 / - | 1,024 / 1,024 | Patch / Patch |
| Large | LLM4TS / TSFMs | 28 / 28 | Qwen-3 / - | 2,048 / 2,048 | Patch / Patch |

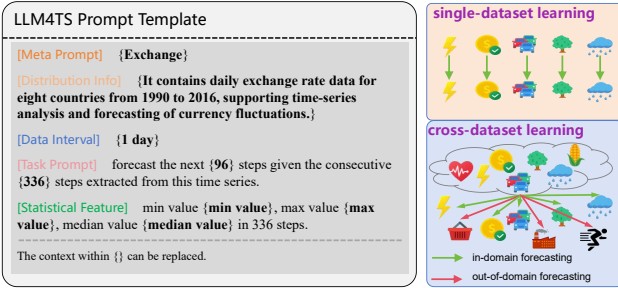

*Figure 2.* (**Left**) The prompt consists of five subtypes, illustrated here using Exchange dataset as an example. (**Right**) In single-dataset learning, the model is trained on individual datasets, while in cross-dataset learning, a many-to-many scheme is adopted to enhance the model's generalization across datasets.

model family, only positional encoding and layer normalization are fine-tuned to avoid catastrophic forgetting (Zhou et al., 2023). We selected lightweight GPT-2 (Radford et al., 2018) and more powerful Qwen-3 (Yang et al., 2025) as the LLM4TS backbones. For the TSFMs model family, they are trained end-to-end with full-parameter. A detailed schematic of the overall architecture is provided in Fig. 1.

**Model Size.** We investigate four distinct model scales: (i)Tiny, (ii)Small, (iii)Base, and (iv)Large. As detailed in Tab. 2, we control the configurations for both the LLM4TS and TSFMs families at each scale, such as the number of layers and backbone choice. This ensures that their learnable parameter counts are approximately comparable within each corresponding size, facilitating a fair comparison.

**TS dataset.** As shown in Tab. 3, the nine selected distributions vary substantially in scale, ranging from 60K to 15M observations and totaling approximately 33M records. These TS datasets are derived from nine real-world data distributions, deliberately chosen to span diverse temporal characteristics, including varying degrees of seasonality, trend, and noise.

**Learning Strategies.** We employ two strategies: single-dataset and cross-dataset learning strategy, shown in Fig. 2. In the former, each model is trained and evaluated on a single dataset (e.g., ETTh1) to measure in-domain performance. In the latter, models are trained on the combined training sets and share weights across the test sets of all datasets (Chang et al., 2025). We follow FSCA (Hu et al., 2025) for dataset

*Table 3.* Variates and observations of datasets, collected from multiple real-world scenarios (Liu et al., 2024b).

| ETTh1 | ETTh2 | ETTm1 | ETTm2 | ECL | Exchange | Solar | Weather | Traffic |
|---|---|---|---|---|---|---|---|---|
| 7 | 7 | 7 | 7 | 321 | 8 | 137 | 21 | 862 |
| 121,940 | 121,940 | 487,760 | 487,760 | 8,443,584 | 60,704 | 7,200,720 | 1,106,616 | 15,122,928 |

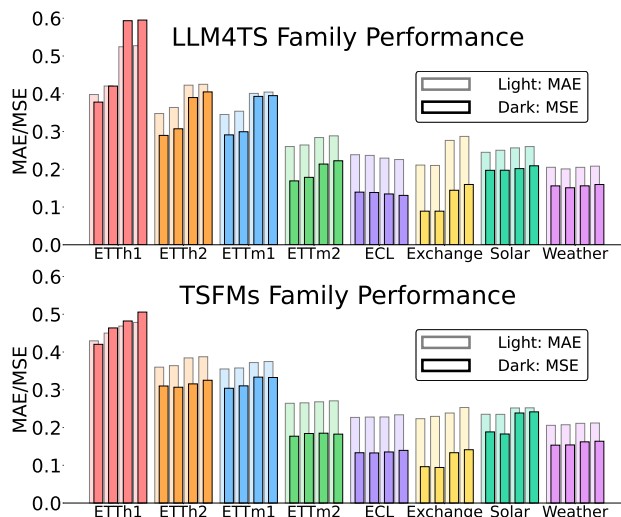

*Figure 3.* Average performance across different model scales and data distribution. (**Upper**) Metrics of the LLM4TS family. (**Lower**) Metrics of the TSFMs family. Lighter bars is MAE, and darker bars is MSE. For each dataset, the 4 bars from *"left to right"* represent *"Tiny to Large"*. The scaling paradox appears across most data distributions and is more pronounced in LLM4TS.

partitioning, using a input of 336 to forecast a horizon of 96. In the single-dataset learning setting, models are evaluated on the same distribution to measure in-domain performance. cross-dataset learning additionally tests on on eight new distributions (NN5, PDB, Sceaux, Smart, Spanish, Sunspot Rain, US Births, and Wind Power) to assess out-of-domain generalization (Goswami et al., 2024).

To structure our investigation, we organize the analysis of these five factors into three key Research Questions (RQs). RQ1 (Sec. 3.2) addresses the most fundamental factors, model architecture and model size, to determine if the scaling paradox is universal. RQ2 (Sec. 3.3) focuses on examining how variations in data volume within the same distribution influence the observed paradox. Finally, RQ3 (Sec. 3.4) investigates how the interplay between data distribution and learning strategies (particularly cross-dataset learning) influences the emergence of the paradox.

### 3.2. Examining the Universality of the Scaling Paradox (RQ1)

**Rationale & Setup.** Empirically, regardless of the large-scale TS models paradigm, a larger model enhances performance, whether strength comes from powerful prior knowledge endowed by LLMs in LLM4TS, or from the fully learnable capacity of TSFMs (Chowdhery et al., 2023; Isik et al., 2025). However, in the field of TS, this theory does not seem to hold. We adopt a single-dataset learning strategy to evaluate model performance across eight datasets with different data volumes, with a particular focus on investigating whether the scaling paradox is consistently observed across the two architectures within model family.

**Results.** Fig. 3 shows that MAE & MSE do not significant decrease as the model scales up. Interestingly, across most datasets, larger models tend to underperform, irrespective of the dataset size, be it the large-scale ECL or the small ETTh. It means that neither the prior knowledge in bigger LLM4TS nor the stronger parameterization in larger TSFMs leads to better performance.

> **Takeaways 1:** Scaling paradox is a pervasive phenomenon in TS modeling, consistently observed across varying data distributions and volumes, and appears to be more pronounced in LLM4TS family.

### 3.3. Impact of Data Scale on the Scaling Paradox (RQ2)

**Rationale & Setup.** Across almost all datasets, despite large variations in distribution and data volume (from 60K to 15M), model scaling yields no significant performance gains (Fig. 3). In this section, so we investigate how data volume relates to the scaling paradox under a consistent distribution. We conduct experiments using two scales from each family: Small and Large for LLM4TS, and Tiny and Large for TSFMs. We use eight datasets and single-dataset learning as in Sec. 3.2, while varying the training data ratio from 20% to 100% with fixed validation and test sets.

**Results.** As the data volume increases from 20%, MAE and MSE exhibit a downward trend. However, larger models still underperform or perform comparably to their smaller counterparts, and the MSE gap between them does not significantly narrow, as shown in Fig. 4. It suggests that increasing data volumes helps to improve the performance of both small and large models, but it does not alleviate the scaling paradox.

> **Takeaways 2:** Increasing the training data improves overall performance but brings no significant relief to the scaling paradox.

### 3.4. Impact of Data Distribution Diversity and Learning Strategies on the Scaling Paradox (RQ3)

**Rationale & Setup.** Training on a narrow single dataset, can cause the model to overfit to specific temporal resolu-

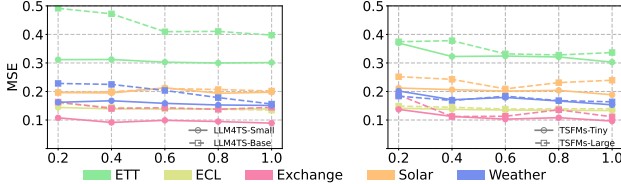

*Figure 4.* MSE under data ratios from 20% to 100%. (**Left**) MSE of LLM4TS at *Small* and *Base* scales. (**Right**) MSE of TSFMs at the *Tiny* and *Large* scales. In the two subfigures, the MSE of large and small models decreases as ratio increases, but the MSE gap does not consistently narrow.

*Table 4.* Performance under single-& cross-dataset learning. ▲ denotes increasing errors with larger model, while ▼ indicates decreasing errors. ★ marks the best performance within the LLM4TS family, and ★ represents the best performance within the TSFMs.

| strategy | Single-dataset learning | | | | | |
|---|---|---|---|---|---|---|
| **Models** | **LLM4TS-** | | | **TSFMs-** | | |
| | Small | Base | Large | Small | Base | Large |
| **MAE** | 0.2875★ | 0.3250 | 0.3283 | 0.2921 | 0.3028 | 0.3078 |
| **MSE** | 0.2226★ | 0.2784 | 0.2846 | 0.2286 | 0.2482 | 0.2540 |
| **Trend** | | ▲ | | | ▲ | |
| strategy | Cross-dataset learning | | | | | |
| **Models** | **LLM4TS-** | | | **TSFMs-** | | |
| | Small-C | Base-C | Large-C | Small-C | Base-C | Large-C |
| **MAE** | 0.2986 | 0.2894 | 0.2875 | 0.2872★ | 0.2986 | 0.3006 |
| **MSE** | 0.2367 | 0.2263 | 0.2233 | 0.2257★ | 0.2496 | 0.2543 |
| **Trend** | | ▼ | | | ▲ | |

tions and limited semantic patterns (Woo et al., 2024; Huang et al., 2025). Such constraints hinder models from fully realizing their generalization potential. We adopt mixed-distribution learning to increase data diversity and scale, and analyze their relationship with model scaling.

We aggregated the training sets listed in Tab. 3 and additionally collected 32 datasets, resulting in a total of 41 datasets for cross-dataset learning, resulting in six models: LLM4TS-(Small, Base, Large)-C and TSFMs-(Small, Base, Large)-C. During evaluation, each model is assessed on each individual test set. In this setting, the data are not only more diverse in distribution, sourced from eight major domains, but also comprise up to 6B observations. After model evaluation, we obtained in-domain (as shown in Tab. 4) and out-of-domain (as shown in Fig. 5) results, and compared the in-domain results with those presented in Sec. 3.2.

**Results.** In Tab. 4, cross-dataset learning improves the in-domain performance of larger models; however, somewhat surprisingly, the best performance still falls slightly below that of ★. In contrast, scaling paradox in TSFMs remains pronounced, and the smallest model achieves the best results ★. In addiction, out-of-domain evaluations fail to gain improvements from larger scales, shown in Fig. 5.

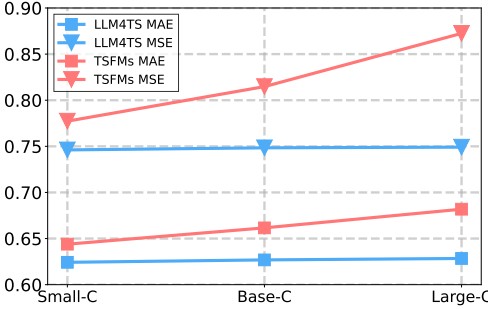

*Figure 5.* Average errors of three scales of LLM4TS and TSFMs models on eight out-of-domain datasets.

> **Takeaways 3:** In in-domain, larger LLM4TS leverage LLMs knowledge but still underperform smaller models trained on single datasets. For TSFMs, mixed learning improves performance at the same scale, but scaling paradox persists. In out-of-domain testing, larger models show no advantage.

## 4. The Few Layers Dominate the Majority

Although larger models contain more layers, they do not yield better performance. As the previous analyses in Sec. 3 have shown this phenomenon is not caused by other factors such as data or training strategies. Therefore, we propose possible hypotheses: *"Not all layers contribute effectively to the prediction process (hypothesis 1)"* and *"Retaining a few true executors can rival the full (hypothesis 2)"*. Experimental results confirm the proposed hypothesis and motivate a new approach for identifying and pruning redundant layers.

### 4.1. Not All Layers Contribute to the Final Predictions

**Inter-layer Representations.** We use Euclidean distance to measure absolute vector differences and reflects overall scaling variations in a geometrical way. Let $\boldsymbol{H}^{l-1}$ and $\boldsymbol{H}^l$ denote the input and output of $l$-th layer. The directional shifts are measured using cosine similarity due to its scale invariance (Kornblith et al., 2019) and are presented below:

$$Dist^l = \|\boldsymbol{H}^l - \boldsymbol{H}^{l-1}\|_2, \quad Sim^l = \frac{\langle \boldsymbol{H}^l, \boldsymbol{H}^{l-1} \rangle}{\|\boldsymbol{H}^l\|_2 \|\boldsymbol{H}^{l-1}\|_2} \quad (1)$$

**Intra-layer Representations.** Each attention head can be regarded as an independent relational learner (Liu et al., 2021), parameterized by distinct projection matrices. Given the embeddings $\boldsymbol{X}^l \in \mathbb{R}^{N \times d_{\text{model}}}$ at layer $l$, the $i$-th head generates its attention weights: $\boldsymbol{A}_i^l = \text{softmax}\left(\boldsymbol{Q}_i^l \boldsymbol{K}_i^{l\top}/\sqrt{d}\right)$, We measure the average pairwise similarity across all head attentions in layer $l$. A lower $\bar{s}^{(l)}$ indicates higher functional diversity among heads, re-

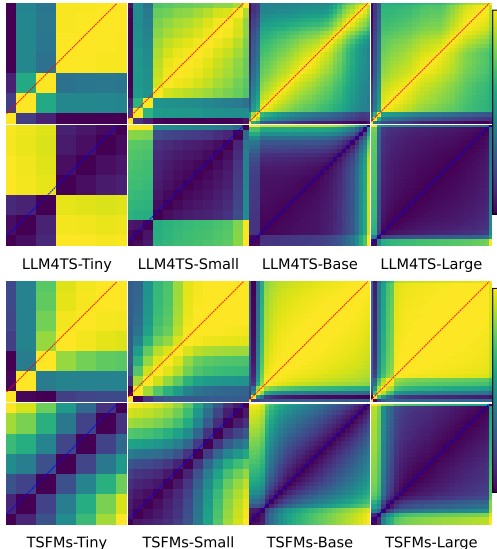

Figure 6. Inter-layer similarity (row 1) and Euclidean distance (row 2). Across eight models of different scales and architectures, inter-layer exhibit high similarity and low Euclidean distance.

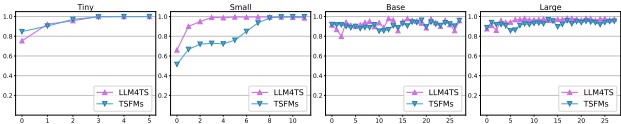

*Figure 7.* Average inter-layer similarity across all head attentions.

flecting that each head captures distinct aspects of the TS representations. $\bar{s}^l$ be defined as:

$$\bar{s}^l = \frac{2}{H(H-1)} \sum_{1 \leq i < j \leq H} \text{sim}\left(\boldsymbol{A}_i^l, \boldsymbol{A}_j^l\right). \quad (2)$$

**Evidence 1.** By analyzing the representations of the eight trained models during the inference process, we find that only a few layers contribute substantial representation shifts, while many layers function redundantly as shown in Fig. 6. This layer-wise redundancy is also observed in cross-dataset learning. Our experiments shows even with increased training data and greater distribution diversity, larger models still exhibit substantial redundancy across most layers, which helps explain why scaling up model size does not always lead better.

**Evidence 2.** Eight model from two families exhibit similar trends in intra-layer representations, as shown in Fig. 7. For models with smaller backbones (6 or 12 layers), heads in the shallow layers primarily allocate their attention to distinct semantic patterns. However, in the middle and deeper layers, $\bar{s}^l$ approaches 1, with most heads learning within the similar subspace. For models with larger backbones (28 layers), heads across all layers almost exhibit high similarity.

**Conclusion:** Only a small fraction of layers actively contribute to representation learning, and the vast majority remain largely passive spectators.

### 4.2. Retaining a Few True Executors Can Rival the Full

**Rationale.** Building on the above, we fine only a small subset making the primary contribution. Then, if only a few seemingly more influential layers are retained while pruning "less important" ones, would the model still exhibit good performance? This motivates the need for a method to assess each layer's contribution and identify critical layers.

**Critical Layer Identification.** In networks, each layer builds upon its predecessors, refining them toward higher-level features. Therefore, static properties of an individual layer may not fully reflect its contribution to last predictions. Considering the decaying influence of preceding layers on deeper layers, importance score is: $I^l = \mathbf{1}\{l \in \text{Top}\} \cdot \left(1 - R^l\right) \cdot \left(1 - \bar{s}^l\right), R^l = \sum_{k=1}^{K} w_k \cdot Sim\left(\boldsymbol{H}^l, \boldsymbol{H}^{l-k}\right).$

$\alpha = 0.5$ denotes the decay factor, $w_k = \alpha^k / \sum_{i=1}^{K} \alpha^i$ the normalized decaying weight, $\bar{s}^l$ is defined in eq. 2 and $K$ is the number of preceding layers. We use the multiplicative form $(1 - R^l) \cdot (1 - \bar{s}^l)$ instead of a weighted sum. This introduces a nonlinear gating effect that downweights redundant or unstable layers, assigning high importance only to layers that are distinctive and stable. A weighted sum could let one factor offset the other, hiding their joint effect in hierarchical feature abstraction. $85\%$ specifies the top percentile of layers ranked by $Dist^l$. By ranking all layers according to $I^l$ in descending order, those contributing most can be identified. The layers are retained if they satisfy either of the two criteria: being within the top $1/2$ or having a cumulative score proportion $> 90\%$, with the latter given higher priority.

**Method Evaluation.** We conduct experiments across 12 models and 9 data distributions, and visualize the average distribution of layer importance over the validation sets of different datasets, as shown in Fig. 14. Based on the importance ranking, we prune redundant layers and fine-tune the pruned model on the training sets to realign it with the data distribution. We evaluate the pruned and original models in in-domain and out-of-domain forecasting scenarios, considering prediction errors and inference efficiency.

**Results.** MAE and MSE are reported as the averages over all datasets. Efficiency is defined as $T_{\text{original model}} / T_{\text{pruned model}}$, and $T$ is inference time. All experiments are conducted under identical settings. Main results are reported in Tab. 5. Results show that the pruned models (minimum retained proportion $\approx \mathbf{20\%}$.) achieve an average of 2× (up to **2.7×**) inference speedup while almost outperforming the original models. The maximum perfor-

mance gain reaches **12%**, but the largest loss is only **0.6%**, which is negligible.

*Table 5.* Comparison of performance and efficiency between the original and pruned models. "ID" and "OOD" refer to in-domain and out-of-domain, respectively. OOD evaluation is not applicable for single-dataset learning, denoted as "–". Performance gain and loss denote improved and degraded metrics.

| Model | MAE-ID | | MSE-ID | | MAE-OOD | | MSE-OOD | | Efficiency↑ | Critical layer ratio |
|---|---|---|---|---|---|---|---|---|---|---|
| | Pruned | Original | Pruned | Original | Pruned | Original | Pruned | Original | | $L_{\text{pruned model}}/L_{\text{original model}}$ |
| **Single-dataset learning** | | | | | | | | | | |
| LLM4TS-Small | 0.283 | 0.288 | 0.218 | 0.223 | – | – | – | – | **2.7** | 28% |
| LLM4TS-Base | 0.293 | 0.325 | 0.226 | 0.278 | – | – | – | – | 2.2 | 23% |
| LLM4TS-Large | 0.283 | 0.328 | 0.219 | 0.285 | – | – | – | – | 2.4 | **21%** |
| TSFMs-Small | 0.287 | 0.292 | 0.221 | 0.227 | – | – | – | – | 2.5 | 24% |
| TSFMs-Base | 0.284 | 0.303 | 0.220 | 0.248 | – | – | – | – | **2.7** | **21%** |
| TSFMs-Large | 0.290 | 0.308 | 0.231 | 0.254 | – | – | – | – | **2.7** | 23% |
| **Cross-dataset learning** | | | | | | | | | | |
| LLM4TS-Small-C | 0.298 | 0.299 | 0.233 | 0.237 | 0.625 | 0.624 | 0.740 | 0.746 | 2.2 | 33% |
| LLM4TS-Base-C | 0.288 | 0.289 | 0.225 | 0.226 | 0.618 | 0.627 | 0.738 | 0.748 | 2.1 | 36% |
| LLM4TS-Large-C | 0.286 | 0.288 | 0.226 | 0.223 | 0.620 | 0.628 | 0.727 | 0.749 | 2.2 | 25% |
| TSFMs-Small-C | 0.286 | 0.287 | 0.227 | 0.226 | 0.635 | 0.644 | 0.782 | 0.777 | 1.9 | 33% |
| TSFMs-Base-C | 0.293 | 0.299 | 0.247 | 0.250 | 0.654 | 0.662 | 0.773 | 0.815 | 2.5 | 29% |
| TSFMs-Large-C | 0.282 | 0.301 | 0.224 | 0.254 | 0.659 | 0.682 | 0.821 | 0.872 | **2.7** | 25% |

> **Conclusion:** Retaining only the critical layers, followed by fine-tuning, can preserve or even improve forecasting accuracy, while substantially inference latency.

# 5. Further Investigation of Our Method

In the previous section, our method demonstrated strong effectiveness when applied to model families. In Sec. 5.1, we extend this method to other model architectures to validate its transferability. Sec. 5.2 investigates 3 configurations, keeping all layers, keeping only a few important layers, and randomly keep same number of layers to assess their impact.

## 5.1. Method Scalability

**Models and Datasets.** Besides the datasets listed in Tab. 3, four PEMS subsets are included for broader coverage, with the splits following (Liu et al., 2024b). We extend the method to four LLM4TS models: FSCA(Hu et al., 2025), CALF(Liu et al., 2025b), TIME-LLM(G) (GPT-2 as backbone)(Jin et al., 2024), and OFA(Zhou et al., 2023), and four TSFMs models: SUNDIAL$_{\text{Large}}$(Liu et al., 2025c), CHRONOS$_{\text{Base}}$(Ansari et al., 2024), MOIRAI$_{\text{Large}}$(Woo et al., 2024) and TIMESFM(Das et al., 2024). With considerable heterogeneity in architecture, scale, temporal encoding method and paradigm, these models provide ideal conditions for assessing the transferability of our method, as shown in Tab. 12. We start from the complete GPT-2, rather than truncating the first few layers as done in FSCA, CALF, and OFA. Note: The four TSFMs are fine-tuned on downstream datasets to strengthen generalization and ensure comparable parameter distributions before & after pruning, following (Zhao et al., 2026).

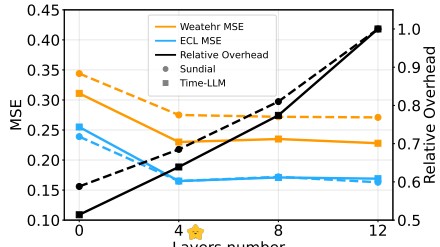

*Figure 8.* Overhead–Accuracy Trade-off.

**Results.** As shown in Tab. 6, for the four LLM4TS models, we retain only one-third of the layers instead of the entire LLM, resulting in a 1.5× speedup. This pruning strategy surpasses the baseline truncation method in over 80% of scenarios, with only a minor performance drop (below 1% on average) in rare cases. As shown in Tab. 7, for the four TSFMs models, we retain only about 26.9% of the layers on average, achieving over a 2.7× speedup. By preserving only the key layers, the performance remains virtually lossless. For various famous LLM4TS and TSFMs in the TS domain, our method demonstrates that retaining only the critical layers not only yields a more lightweight model, but also achieves comparable or even superior forecasting accuracy.

## 5.2. Less Is More: Full LLMs Are Not Always Beneficial

To balance the use of LLMs' prior knowledge and their vast parameter scale, FSCA, OFA, and CALF adopt a coarse-grained strategy by directly truncating the early layers. However, such pruning is not only ineffective in preserving the complete semantic of LLM, but also lacks precise control over the number of retained layers. Our method provides a more principled approach, leading to better performance.

In this section, our approach both reduces semantic interference introduced by redundant parameters, and outperforms coarse-grained pruning. Specifically, we explore the impact of 3 layer retention strategies on predictions. **Pruned**: our pruning method; **Full**: entire LLM used as backbone without pruning; **Random**: An equal number of layers to those retained by our pruning method are selected, but their positions are randomly chosen. Tab. 8 shows that our models not only outperform the models using the full backbone, but also surpass the randomly selected layers of equal number, regardless of their positions.

## 5.3. Ablations

**Pruning Without Fine-tuning.** In the previous experiments, we fine-tune the model after removing redundant layers. Here, we analyze the negative effects when fine-tuning is omitted. As shown in Tab. 9, omitting fine-tuning leads to a failure in bridging the gap between the pruned structure and the parameter distribution. Hence, fine-tuning

*Table 6.* Pre- and post-pruning metrics in four LLM4TS. Results are averaged across horizons $\in \{96, 192, 336, 720\}$. Original: model from original paper. Performance gain and loss denote improved and degraded metrics.

| Models | LLM4TS | | | | | | | | | | | | | | | |
|---|---|---|---|---|---|---|---|---|---|---|---|---|---|---|---|---|
| | FSCA (2025) | | | | CALF (2025b) | | | | Time-LLM(G) (2024) | | | | OFA (2023) | | | |
| | Pruned | | Original | | Pruned | | Original | | Pruned | | Original | | Pruned | | Original | |
| Metric | MAE | MSE | MAE | MSE | MAE | MSE | MAE | MSE | MAE | MSE | MAE | MSE | MAE | MSE | MAE | MSE |
| ETTh1 | 0.443 | 0.426 | 0.444 | 0.430 | 0.431 | 0.436 | 0.434 | 0.446 | 0.446 | 0.431 | 0.459 | 0.448 | 0.435 | 0.429 | 0.434 | 0.430 |
| ETTh2 | 0.390 | 0.349 | 0.390 | 0.348 | 0.391 | 0.362 | 0.395 | 0.373 | 0.408 | 0.368 | 0.410 | 0.370 | 0.403 | 0.366 | 0.403 | 0.366 |
| ETTm1 | 0.385 | 0.350 | 0.387 | 0.352 | 0.384 | 0.387 | 0.387 | 0.391 | 0.390 | 0.358 | 0.389 | 0.359 | 0.388 | 0.357 | 0.385 | 0.355 |
| ETTm2 | 0.319 | 0.258 | 0.320 | 0.259 | 0.320 | 0.268 | 0.318 | 0.277 | 0.327 | 0.268 | 0.330 | 0.274 | 0.325 | 0.265 | 0.325 | 0.265 |
| ECL | 0.266 | 0.165 | 0.267 | 0.166 | 0.269 | 0.184 | 0.272 | 0.189 | 0.263 | 0.163 | 0.271 | 0.169 | 0.259 | 0.166 | 0.265 | 0.169 |
| Weather | 0.269 | 0.233 | 0.268 | 0.229 | 0.271 | 0.255 | 0.277 | 0.259 | 0.266 | 0.228 | 0.266 | 0.228 | 0.257 | 0.230 | 0.268 | 0.232 |
| Exchange | 0.434 | 0.436 | 0.449 | 0.450 | 0.413 | 0.378 | 0.408 | 0.370 | 0.448 | 0.436 | 0.448 | 0.441 | 0.408 | 0.384 | 0.428 | 0.410 |
| Solar | 0.264 | 0.198 | 0.274 | 0.210 | 0.260 | 0.227 | 0.268 | 0.251 | 0.257 | 0.190 | 0.257 | 0.189 | 0.264 | 0.211 | 0.276 | 0.216 |
| Traffic | 0.277 | 0.390 | 0.284 | 0.397 | 0.284 | 0.424 | 0.286 | 0.465 | 0.279 | 0.395 | 0.284 | 0.398 | 0.275 | 0.404 | 0.298 | 0.422 |
| PEMS03 | 0.264 | 0.171 | 0.270 | 0.171 | 0.401 | 0.363 | 0.406 | 0.373 | 0.282 | 0.183 | 0.284 | 0.185 | 0.268 | 0.178 | 0.281 | 0.183 |
| PEMS04 | 0.346 | 0.458 | 0.358 | 0.463 | 0.508 | 0.772 | 0.513 | 0.778 | 0.376 | 0.482 | 0.379 | 0.484 | 0.362 | 0.477 | 0.375 | 0.488 |
| PEMS07 | 0.221 | 0.115 | 0.242 | 0.130 | 0.394 | 0.384 | 0.405 | 0.393 | 0.250 | 0.137 | 0.253 | 0.142 | 0.235 | 0.136 | 0.252 | 0.145 |
| PEMS08 | 0.345 | 0.510 | 0.353 | 0.509 | 0.488 | 0.770 | 0.506 | 0.837 | 0.377 | 0.527 | 0.386 | 0.544 | 0.359 | 0.510 | 0.387 | 0.557 |
| | **WR: 77%, PR: 69.7%, SP: 1.53x** | | | | **WR: 88%, PR: 63.2%, SP: 1.48x** | | | | **WR: 77%, PR: 63.2%, SP: 1.57x** | | | | **WR: 73%, PR: 60.2%, SP: 1.51x** | | | |

\* WR: Win Rate; PR: Average Pruning Layer Ratio; SP: Average Speedup Ratio.

*Table 7.* Comparison of pre- and post-pruning metrics in four TSFMs. Results are averaged across horizons $\in \{96, 192, 336, 720\}$. FT: fine-tuning on new data distributions. Pruning + FT: Prune-then-finetune.

| Models | TSFMs | | | | | | | | | | | | | | | |
|---|---|---|---|---|---|---|---|---|---|---|---|---|---|---|---|---|
| | Sundial$_{Large}$ (2025c) | | | | Chronos$_{Base}$ (2024) | | | | Moirai$_{Large}$ (2024) | | | | TimesFM (2024) | | | |
| | Pruning + FT | | FT | | Pruning + FT | | FT | | Pruning + FT | | FT | | Pruning + FT | | FT | |
| Metric | MAE | MSE | MAE | MSE | MAE | MSE | MAE | MSE | MAE | MSE | MAE | MSE | MAE | MSE | MAE | MSE |
| ETTh1 | 0.408 | 0.387 | 0.418 | 0.398 | 0.405 | 0.416 | 0.411 | 0.424 | 0.422 | 0.425 | 0.429 | 0.433 | 0.411 | 0.410 | 0.416 | 0.415 |
| ETTh2 | 0.382 | 0.327 | 0.390 | 0.334 | 0.366 | 0.337 | 0.370 | 0.344 | 0.392 | 0.348 | 0.391 | 0.351 | 0.369 | 0.332 | 0.376 | 0.337 |
| ETTm1 | 0.367 | 0.329 | 0.374 | 0.334 | 0.356 | 0.347 | 0.358 | 0.354 | 0.374 | 0.352 | 0.381 | 0.357 | 0.372 | 0.366 | 0.379 | 0.380 |
| ETTm2 | 0.315 | 0.250 | 0.315 | 0.255 | 0.298 | 0.255 | 302 | 0.268 | 0.313 | 0.260 | 0.317 | 0.263 | 0.301 | 0.254 | 0.307 | 0.259 |
| ECL | 0.260 | 0.161 | 0.262 | 0.163 | 0.230 | 0.150 | 0.244 | 0.156 | 0.269 | 0.174 | 0.274 | 0.179 | - | - | - | - |
| Weather | 0.227 | 0.265 | 0.233 | 0.271 | 0.246 | 0.222 | 0.267 | 0.257 | 0.282 | 0.238 | 0.292 | 0.253 | - | - | - | - |
| | **WR: 92%, PR: 72.9%, SP: 2.87x** | | | | **WR: 100%, PR: 71.3%, SP: 2.64x** | | | | **WR: 92%, PR: 67.1%, SP: 2.33x** | | | | **WR: 100%, PR: 81.2%, SP: 3.09x** | | | |

*Datasets that have already been involved in pre-training are excluded from evaluation, and are denoted by a dash ("–").

*Table 8.* MSE of different retained layers strategy. Our pruning method achieves the lowest MSE, outperforming both the Full and Random schemes. Horizons $\in \{96, 192, 336, 720\}$.

| Models | FSCA | | | CALF | | | OFA | | |
|---|---|---|---|---|---|---|---|---|---|
| | Pruned | Full | Random | Pruned | Full | Random | Pruned | Full | Random |
| ETTh1 | 0.426 | 0.442 | 0.432 | 0.436 | 0.445 | 0.461 | 0.429 | 0.461 | 0.451 |
| ETTh2 | 0.349 | 0.367 | 0.355 | 0.362 | 0.374 | 0.374 | 0.366 | 0.379 | 0.384 |
| ETTm1 | 0.350 | 0.368 | 0.360 | 0.387 | 0.392 | 0.397 | 0.357 | 0.365 | 0.366 |
| ETTm2 | 0.258 | 0.267 | 0.258 | 0.268 | 0.280 | 0.279 | 0.265 | 0.287 | 0.282 |
| Weather | 0.233 | 0.237 | 0.244 | 0.255 | 0.267 | 0.278 | 0.230 | 0.248 | 0.231 |
| avg. | 0.323 | 0.336 | 0.330 | 0.342 | 0.352 | 0.358 | 0.329 | 0.348 | 0.343 |

*Table 9.* Results of pruning w & w/o fine-tuning. **Vanilla** refers to fine-tuning after pruning, while **w/o** indicates that fine-tuning is omitted. △ indicates the difference.

| Dataset | Metric | OFA | | | Sundial$_{Large}$ | | |
|---|---|---|---|---|---|---|---|
| | | vanilla | w/o | △ | vanilla | w/o | △ |
| ETTh1 | MAE | 0.435 | 0.554 | 0.121 | 0.408 | 0.512 | 0.104 |
| | MSE | 0.429 | 0.627 | 0.198 | 0.387 | 0.455 | 0.068 |
| ECL | MAE | 0.259 | 0.318 | 0.059 | 0.260 | 0.334 | 0.074 |
| | MSE | 0.166 | 0.223 | 0.057 | 0.161 | 0.217 | 0.056 |

after pruning is indispensable.

**Trade-off Between Efficiency and Accuracy.** We retain different layer numbers in descending order of importance to explore the trade-off. Relative overhead is defined as $T_{\text{pruned model}}/T_{\text{original model}}$, and $T$ is inference time. Fig. 8 shows just keep top 4 critical layers perform well, yet eliminating all layers results in a notably worse MSE. Retaining

1/3 of the layers achieves an optimal balance, as the forecasting performance is almost saturated while requiring merely 65% of the inference time compared to the full model.

# 6. Related Work

**Large Language Models for Time Series.** Applying fully pre-trained LLMs to TS tasks poses the central challenge

of achieving effective modality alignment. Existing efforts can be broadly categorized into two directions: TS embedding and latent space alignment (Zhang et al., 2024; Pan et al., 2024; Liu et al., 2025b). TS embedding maps raw TS into representations that are more compatible with the pre-trained LLMs space, simplifying knowledge transfer (Jin et al., 2024; Liu et al., 2024a). When combined with prompt-based mechanisms, the strategy leverages intrinsic capacity of LLMs to process TS tasks in a form that preserves generalization abilities (Liu et al., 2025a). By contrast, latent-space alignment seeks to directly bridge representational gap (Zhou et al., 2023; Hu et al., 2025; Liu et al., 2025b). It is achieved through joint training or contrastive objectives that promote cross-modal alignment, improving feature sharing and transferability. One line of work retains original structure of LLMs, mapping TS to input layer (Jin et al., 2024). It preserves pretrained knowledge and avoids retraining overhead, but may suffer from parameter incompatibility, reducing efficiency and adaptability. In contrast, another line of work employs only a few shallow layers, which reduces computational complexity and parameter scale, improve inference efficiency and alleviate deployment costs (Pan et al., 2024; Chang et al., 2025). However, it's drawback lies in the loss of pretrained knowledge, diminishing ability to capture local patterns and long-range dependencies.

**Time Series Foundation Models.**  TSFMs aim to construct large-scale, general-purpose models for TS tasks by learning on massive heterogeneous datasets, enabling generalized capabilities across tasks (Ansari et al., 2024; Das et al., 2024; Liu et al., 2025c). TSFMs show strong zero-shot potential, enabling accurate predictions across unseen domains, variable types, and temporal granularities with less reliance on task-specific training data. To better accommodate inherent properties of TS (Liu et al., 2024c), TSFMs typically incorporate customized adaptations of Transformer architecture. At the same time, they rely on large-scale, cross-domain pretraining to capture diverse patterns from multi-domains. However, most current TSFMs either directly adopt or only slightly modify LLMs architectures, overlooking fundamental discrepancies between TS and text in terms of statistical properties, dependency structures, and semantic representations (Das et al., 2024; Goswami et al., 2024; Liu et al., 2025c). Capturing the statistical and structural uniqueness of TS remains challenging, while current TSFMs still suffer from limited parameter efficiency.

**Time Series Representation Learning.**  Representation learning is an indispensable research direction in TS analysis. Effective representations can capture important patterns of TS, reduce redundancy, and enhance performance of downstream tasks (Tong et al., 2023a). Among current approaches, multi-scale pyramid structures decompose the original TS into hierarchical features at different resolutions, reducing computational overhead (Liu et al., 2022). Frequency-domain decomposition and frequency-enhanced attention mechanisms, leveraging Fourier or wavelet transforms, optimize seasonal pattern modeling and enable low-complexity representation learning (Zhou et al., 2022). TS can thus be decomposed into seasonal, trend, and residual components to capture multi-scale dynamic patterns (Wu et al., 2021). Next-token prediction employ patch-based inputs, discretization, or modality alignment to pre-train models, enhancing numerical reasoning and inference capabilities (Cao et al., 2024; Xue & Salim, 2023). Multi-task learning further integrates reconstruction and prediction errors to improve robustness and expressiveness of learned representations (Liu et al., 2024a). However, TS model internals remain underexplored, limiting insight into their reasoning, interpretability, and transferability.

**Redundancy Analysis in Large-scale Models.**  Redundancy has been extensively studied in large-scale models across language (Yao et al., 2022; Ma et al., 2023; Men et al., 2025; Zhao et al., 2025) and vision domains (Sung et al., 2024; Fan et al., 2025; Liang et al., 2025; Liu et al., 2026). These studies demonstrate that a substantial amount of parameter redundancy exists within such models, yet observes that performance still scales positively with model size (Kaplan et al., 2020; Aghajanyan et al., 2023; Chen et al., 2024). This phenomenon is commonly attributed to the strong representational capacity and optimization stability brought by overparameterization. In contrast, our findings on large-scale TS models reveal a scaling paradox, where enlarging model size does not necessarily yield better forecasting performance, yet the underlying causes remain largely unexplored.

# 7. Conclusion

In this work, we conduct evaluations of large-scale TS models across architectures, scales, data volumes, distributions, and learning strategies to investigate their effects on the scaling paradox, provide the first in-depth analysis of inter- and intra-layer representations, reveal the phenomenon of few-layer dominance whereby only a small subset of layers are critical, and further propose a critical-layer identification method that preserves forecasting accuracy while improving inference efficiency, with extensive validation on eight representative models confirming its universality. **In future**, we will incorporate the discovered few-layer dominance phenomenon into the architectural design of large-scale TS models, aiming to achieve more efficient and high-performing forecasting systems. **The limitation** is that our analysis is still conducted within mainstream model architectures and training paradigms, while the field is evolving rapidly and new approaches continue to emerge.

## Impact Statement

This paper aims to advance research in the field of machine learning. We believe that this work does not raise specific ethical concerns, nor does it entail direct or immediate societal impacts beyond contributing to academic progress.

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

# A. Theoretical Analysis

We provide a mechanistic analysis explaining why many layers in large-scale TS models converge to near-identity mappings after training, and how this behavior gives rise to the observed scaling paradox. Our analysis follows a structured chain of reasoning: (1) strong temporal autocorrelation constrains representation evolution, (2) constrained evolution induces weak gradients for many layers under residual architectures, and (3) regularized optimization drives such layers toward near-identity mappings.

## A.1. Autocorrelation and Smoothness in TS

Let $\{x_t\}_{t=1}^T$ be a TS and $y$ the forecasting target.

**Assumption 1: Strong temporal autocorrelation.** There exists a finite lag set $\mathcal{K}$ and a constant $\rho \in (0, 1)$ such that for all $k \in \mathcal{K}$, $\mathrm{Corr}(x_t, x_{t-k}) \geq \rho$. Equivalently, the innovation component $\eta_t = x_t - \mathbb{E}[x_t \mid x_{t-1}, \ldots, x_{t-k}]$ has small variance for $k \in \mathcal{K}$. We formalizes the empirical fact that TS inputs exhibit strong redundancy across nearby time steps.

**Assumption 2: Local smoothness of the forecasting map.** Let $u \in \mathbb{R}^d$ denote the model input (e.g., a flattened history window), and let $f(\cdot; \theta)$ denote the model output. There exists a neighborhood $\mathcal{U}$ of typical TS inputs such that the conditional expectation $u \mapsto \mathbb{E}[y \mid u]$ is $L_y$-Lipschitz on $\mathcal{U}$. Due to strong autocorrelation, small perturbations of the input window do not induce abrupt changes in the predictive distribution.

## A.2. Residual Transformer Formulation and Optimization

We abstract a (pre-norm) Transformer as a residual network with layer-wise updates $x_{l+1} = x_l + g_l(x_l; \theta_l)$ for $l = 0, \ldots, L - 1$, where $x_l$ is the hidden representation at layer $l$, $g_l$ denotes the residual transformation, and $\theta_l$ are layer parameters. Training is performed by minimizing the regularized objective:

$$\mathcal{J}(\theta) = \mathbb{E}_{(u,y)\sim\mathcal{P}}[\ell(f(u;\theta), y)] + \frac{\lambda}{2} \sum_{l=0}^{L-1} \|\theta_l\|_2^2,$$

where $\ell$ is a differentiable loss and $\lambda > 0$ is the weight decay coefficient.

**Assumption 3: Zero-output and parameter Lipschitzness.** For each layer $l$ and all typical representations $x$, $g_l(x; 0) = 0$. Moreover, $g_l$ is Lipschitz continuous with respect to $\theta_l$, i.e., there exists $C_l(x) \geq 0$ such that $\|g_l(x; \theta_l)\|_2 \leq C_l(x)\|\theta_l\|_2$.

## A.3. From Weak Gradients to Near-Identity Layers

We first establish that weak marginal influence on the loss implies small parameter norms under regularization.

**Proposition 1: Weak gradients imply small parameter norms.** Let $\theta^\star$ be a local minimizer of $\mathcal{J}(\theta)$. For each layer $l$, $\theta_l^\star = -\frac{1}{\lambda} \nabla_{\theta_l} \mathbb{E}[\ell(f(u;\theta^\star), y)]$. Consequently, $\|\theta_l^\star\|_2 = \frac{1}{\lambda}\|\nabla_{\theta_l} \mathbb{E}[\ell(f(u;\theta^\star), y)]\|_2$.

*Proof.* At a local minimum, the first-order optimality condition gives $\nabla_{\theta_l} \mathcal{J}(\theta^\star) = 0$. Expanding the gradient yields $\nabla_{\theta_l} \mathbb{E}[\ell] + \lambda \theta_l^\star = 0$, from which the result follows.

**Proposition 2: Small parameters imply near-identity mappings.** Under Assumption 3, for any layer $l$ and input $x_l$, $\|x_{l+1} - x_l\|_2 = \|g_l(x_l; \theta_l)\|_2 \leq C_l(x_l)\|\theta_l\|_2$.

*Proof.* By definition, $x_{l+1} - x_l = g_l(x_l; \theta_l)$. The result follows directly from Assumption 3.

## A.4. Why Deep Layers Receive Weak Gradients in TS?

We now explain why many layers satisfy the weak-gradient condition in strongly autocorrelated TS.

**Lemma 1: Gradient decomposition in residual networks.** Let $f(u; \theta) = h(x_L)$ for a differentiable head $h$. Then $\nabla_{\theta_l}\ell(f(u;\theta), y) = J_{\theta_l}g_l(x_l;\theta_l)^\top \nabla_{x_{l+1}}\ell(f(u;\theta), y)$, where $\nabla_{x_{l+1}}\ell(f(u;\theta), y) = \left(\prod_{k=l+1}^{L-1}(I + J_{x_k}g_k)\right)^\top \nabla_{x_L}\ell(h(x_L), y)$.

*Proof.* Both identities follow directly from repeated application of the chain rule.

**Assumption 4: Autocorrelation-induced representational saturation.** There exists a small index set $\mathcal{D}$ of dominant layers, with $|\mathcal{D}| \ll L$, such that for $l \notin \mathcal{D}$, $\|\nabla_{x_{l+1}}\mathbb{E}[\ell(f(u;\theta),y)]\|_2 \leq \gamma_l$, where $\gamma_l$ is small.

**Theorem 1: Layer laziness under strong autocorrelation.** Under Assumption 4, for all $l \notin \mathcal{D}$, $\|\nabla_{\theta_l}\mathbb{E}[\ell(f(u;\theta),y)]\|_2 \leq \|J_{\theta_l}g_l\|_{\mathrm{op}}\gamma_l$. Consequently, such layers satisfy $\|x_{l+1} - x_l\|_2 \leq C_l(x_l)\|J_{\theta_l}g_l\|_{\mathrm{op}}\gamma_l/\lambda$, and converge to near-identity mappings.

*Proof.* By Lemma 1 and Cauchy–Schwarz, $\|\nabla_{\theta_l}\ell\|_2 \leq \|J_{\theta_l}g_l\|_{\mathrm{op}}\|\nabla_{x_{l+1}}\ell\|_2$. Taking expectation and applying Assumption 4 yields the bound. The conclusion follows from Propositions 1 and 2.

### A.5. Connection to the Scaling Paradox

When model depth is increased beyond the small set of dominant layers $\mathcal{D}$, newly added layers fall into the non-dominant regime described above. Such layers are driven toward near-identity mappings, contributing little to effective representation transformation while increasing parameter count and computational cost. This provides a structural explanation for the scaling paradox observed in TS models: scaling depth does not yield proportional performance gains, as additional layers are incentivized to become functionally redundant.

Fig. 9 illustrates the cumulative contribution of residual layers to the learned function, measured by the cumulative residual energy $\mathcal{E}(k)$. For both LLM4TS and TSFMs across different model scales (Tiny, Small, Base, and Large), the curves rise rapidly and reach around 90% of the total residual energy within approximately 20–30% of the layers. This indicates that only a small fraction of layers account for the majority of functional changes, while the remaining layers contribute marginally and behave close to identity mappings. Notably, this concentration effect persists across model scales, suggesting that increasing depth primarily adds functionally redundant layers rather than proportionally increasing effective model capacity.

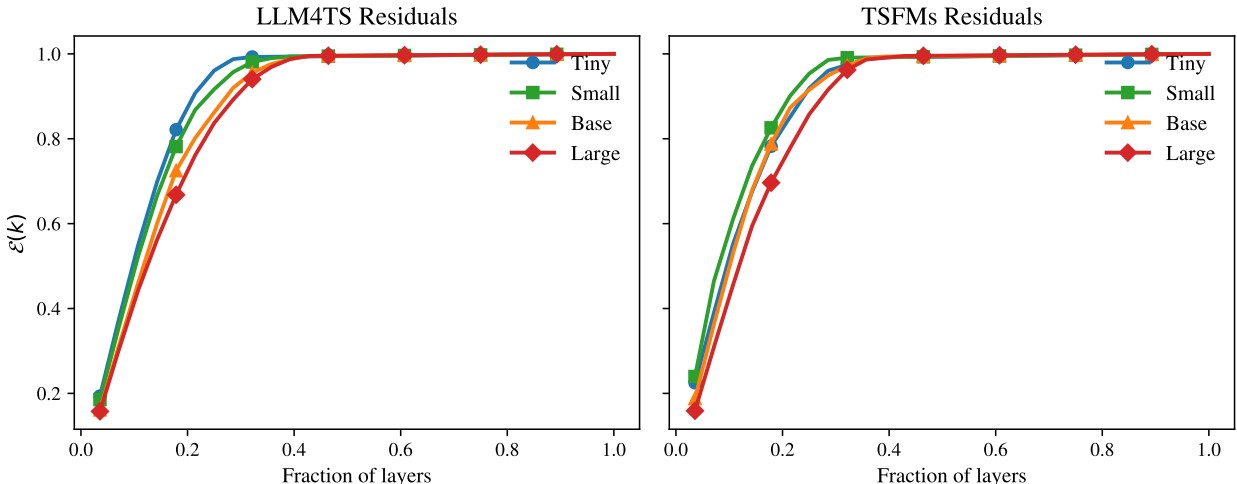

*Figure 9.* Cumulative residual energy across layers for LLM4TS and TSFMs at different model scales. Let $g_l(x_l)$ denote the residual update at layer $l$, and define the residual energy of layer $l$ as $E_l = \mathbb{E}\big[\|g_l(x_l)\|_2^2\big]$, where the expectation is taken over input samples. Layers are sorted in descending order of $E_l$. The curve $\mathcal{E}(k) = \sum_{i=1}^{k} E_{(i)} \big/ \sum_{l=1}^{L} E_l$ denotes the cumulative fraction of residual energy contributed by the top $k$ layers.

# B. Datasets

## B.1. Datasets of Single-Dataset Learning

In the *single-dataset learning* setting, models are trained, validated, and tested on temporally contiguous splits drawn from a single TS distribution. It evaluates the model's in-domain & full-shot learning capability, learn and extrapolate temporal patterns when abundant target-domain data is available during training. Datasets of single-Dataset learning:

**ETT:** It includes four variants: ETTh1 and ETTh2 (hourly) and ETTm1 and ETTm2 (15-minutely). ETT are collected from two distinct regions over approximately two years. Each contains 7 variables, such as oil temperature, load, and equipment status and exhibit strong periodicity, trend drifts, and occasional abrupt shifts, making them standard benchmarks for long-term forecasting.

**ECL:** Hourly electricity consumption records from 321 clients over four years. It features high volatility, heterogeneous load profiles, and complex inter-client correlations, challenging models to capture both global trends and individual behavioral dynamics.

**Exchange:** Daily exchange rates of eight major currencies — Australia, UK, Canada, Switzerland, China, Japan, New Zealand, and Singapore, relative to the US dollar, spanning 16 years. It reflects slow macroeconomic trends, global financial coupling, and rare regime shifts, ideal for evaluating robustness under low-frequency non-stationarity.

**Solar:** 10-minute resolution solar power production from 137 stations in Alabama over one year. Dominated by strong diurnal cycles and weather-induced intermittency, it presents challenges in modeling fine-grained, spatially correlated renewable generation.

**Weather:** 21 meteorological variables, including air temperature, humidity, wind speed, and pressure, are recorded hourly at a German weather station over eight years. Exhibits rich multi-scale dynamics, suitable for evaluating hierarchical temporal modeling.

**Traffic:** It contains hourly road occupancy rates, the percentage of time a road segment is occupied by vehicles, and are recorded across 862 sensors on San Francisco Bay Area freeways over two years. It exhibits complex spatio-temporal correlations, strong weekly seasonality, and sharp rush-hour peaks, making it a challenging benchmark for modeling large-scale, real-world urban dynamics.

**Data Splitting Protocol.** To ensure consistency and avoid temporal leakage, all datasets are partitioned chronologically. For the four ETT datasets, we follow the established 8:4:4 month-wise split: first 8 months for training, next 4 for validation, last 4 for testing. For all other datasets, we use a 7:1:2 ratio by sequence length: 70% training, 10% validation, 20% testing. All variables are standardized using training-set statistics only. No external covariates or calendar features are used, meaning models rely solely on historical observations.

## B.2. Datasets of Cross-Dataset Learning

In the cross-dataset learning setting, models are trained on the union of training sets from multiple distinct TS distributions, validated on the union of validation sets, and finally evaluated separately on each dataset's test set. This protocol evaluates a model's ability to learn universal temporal representations that generalize across domains. Unlike single-dataset learning, this setup tests robustness to distribution shift, and parameter efficiency under heterogeneous data. We extend the dataset collection, resulting in a total of 41 datasets for cross-dataset experiments. The 41 datasets span eight major domains, with their distribution shown in Fig. 10. During training, batches are sampled uniformly across datasets to ensure balanced exposure. At test time, metrics are computed individually per dataset to assess both in-domain and full-shot learning performance. To further evaluate the out-of-domain generalization and zero-shot learning performance of models trained under the cross-dataset protocol, we introduce a held-out evaluation suite comprising eight additional TS datasets. These datasets are never seen during training or validation. Only their test sets are used for final evaluation, providing a strict zero-shot benchmark that measures how well learned temporal representations transfer to completely unseen domains.Datasets of cross-dataset learning:

**NN5:** A collection of 111 daily TS representing sales records of non-food items from an anonymous retail chain in the UK, spanning 2 years. Characterized by intermittent demand, promotional spikes, and calendar-driven seasonality.

**PDB (Protein Data Bank):** Weekly counts of new protein structure submissions to the PDB from 2000 to 2022. Exhibits slow, science-policy-driven growth trends with occasional abrupt shifts due to technological or institutional changes.

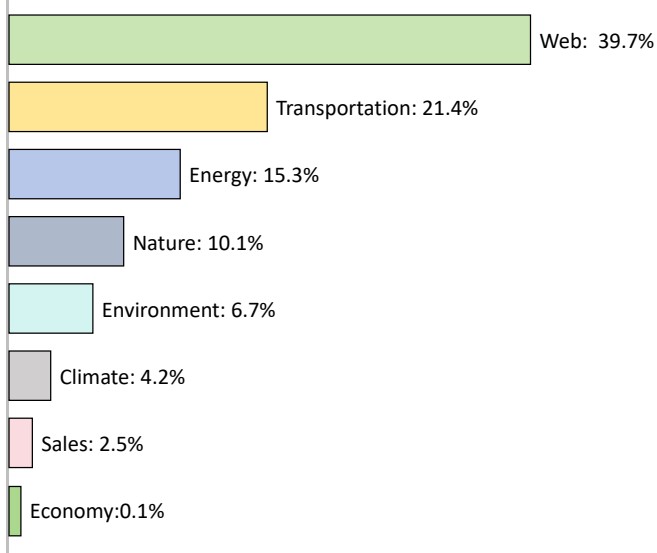

*Figure 10.* Dataset sources.

**Sceaux:** Hourly measurements of temperature, humidity, and pressure recorded at a weather station in Sceaux, France, over 5 years. Features fine-grained meteorological dynamics with strong diurnal and seasonal cycles.

**Smart:** Electricity consumption readings from 30 smart meters in Irish households over 18 months at 30-minute intervals. Captures heterogeneous household behavior, appliance-level patterns, and weather-sensitive usage.

**Spanish:** Daily electricity demand across Spain from 2014 to 2019. Reflects national-scale consumption modulated by economic activity, holidays, temperature, and renewable penetration.

**Sunspot-rain:** A bivariate dataset combining monthly international sunspot numbers and regional rainfall in Eastern Asia from 1850 to 2020. Offers an ultra-long-term, low-frequency view of potential solar-terrestrial coupling.

**USbirths:** Daily counts of births in the United States from 1969 to 1988. Displays strong weekly periodicity, holiday dips, and long-term demographic trends.

**Wind-power:** Hourly wind power generation from 20 turbines in a European wind farm over one year. Highly volatile and non-stationary, governed by weather dynamics and turbine-specific efficiency curves.

**Evaluation Protocol.** Critically, for all eight datasets above, only the test set is used to test and no training or validation samples are exposed to the model at any stage. It ensures a pure zero-shot evaluation: models must forecast these sequences based solely on representations learned from the original nine datasets.

### B.3. Datasets for Baseline Representation Analysis and Pruning Studies

To evaluate the generality of our findings, particularly in representation learning, critical layers localization and model compression, we extend our datasets with 4 additional traffic flow datasets from the PEMS (Performance Measurement System) family: PEMS03, PEMS04, PEMS07, and PEMS08. These datasets provide large-scale,w real-world transportation dynamics and serve as stress tests for structural robustness and efficiency under distribution shift.

**PEMS03:** Hourly traffic occupancy rates recorded by 358 sensors across the San Francisco Bay Area over 3 months in 2018. Exhibits strong spatial correlation and morning/evening rush-hour patterns.

**PEMS04:** Data from 307 sensors in the same region over 3 months in 2018, capturing similar dynamics to PEMS03 but with distinct sensor coverage and flow characteristics.

**PEMS07:** Records from 883 sensors in the Los Angeles metropolitan area over 6 months in 2017. Features higher spatial density and more complex congestion propagation patterns.

**PEMS08:** Measurements from 170 sensors in San Bernardino County over 6 months in 2016. Reflects suburban traffic dynamics with lower density but longer commute corridors.

**Data Splitting Protocol.** All four PEMS datasets follow a 7:1:2, 70% for training, 10% for validation, and 20% for testing. Variables are standardized per dataset using training-set statistics only. No external features are included, ensuring models rely solely on learned temporal-spatial representations. PEMS are not used in cross-dataset learning or zero-shot evaluation.

## B.4. Diverse Data Distributions

Seasonality measures the strength of recurring temporal patterns, Trend quantifies long-term directional changes, Stationarity reflects the stability of statistical properties over time, Transition captures abrupt regime shifts, Shifting characterizes gradual distributional changes, Correlation indicates inter-variable dependency, and Non-Gaussianity measures deviations from Gaussian distributions. As shown in Fig. 11, these metrics reveal the distinct statistical characteristics of the seven datasets, highlighting substantial variability across data distributions. Such diversity provides a comprehensive and challenging testbed for training and evaluating our models.

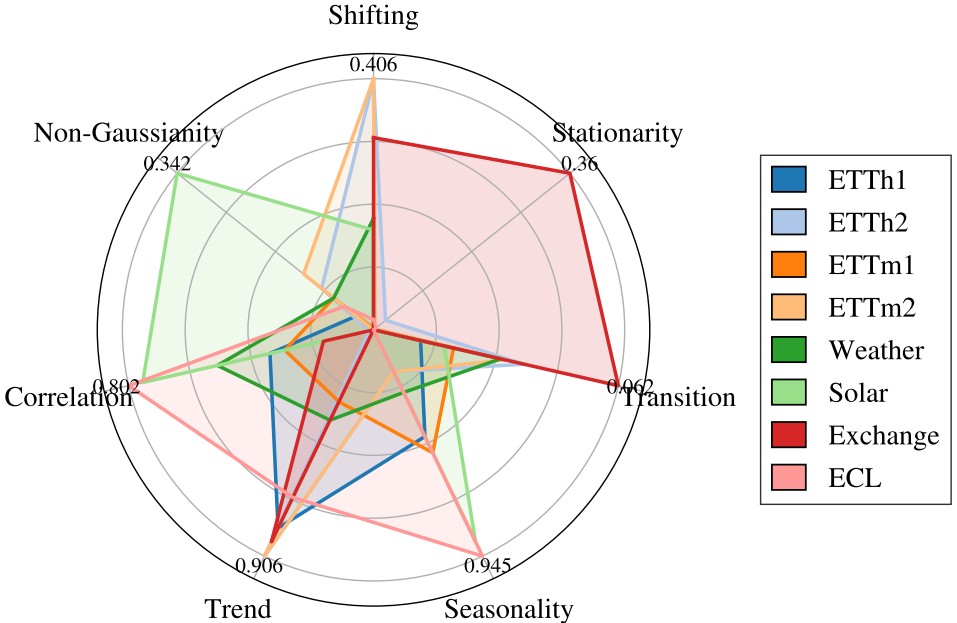

*Figure 11.* Statistical characteristics of datasets, including seasonality, trend, stationarity, transition, shifting, correlation, and non-Gaussianity.

# C. Implementation Details

## C.1. Algorithm: Critical Layer Identification

We present the procedure for critical layer identification in Tab. 10, which details the step-by-step process for computing layer importance scores and ranking layers accordingly.

*Table 10.* Critical layer identification

---

**Algorithm: Critical Layer Identification**

---

**Input:** Layer representations $\boldsymbol{H}^0, \ldots, \boldsymbol{H}^{L-1}$;

   attention weights $\boldsymbol{A}_i^l$ for each head $i$ at layer $l$;

   number of preceding layers $K$ used for temporal comparison.

**Output:** An ordered list of critical layers ranked by importance score.

**for** $l = 0$ **to** $L - 1$ **do**

   *// Representation change across adjacent layers*

   $Dist^l \leftarrow \|\boldsymbol{H}^l - \boldsymbol{H}^{l-1}\|_2$

   *// Attention diversity within the layer*

   $\bar{s}^l \leftarrow \frac{1}{\binom{H}{2}} \sum_{i<j} Sim(\boldsymbol{A}_i^l, \boldsymbol{A}_j^l)$

   *// Weighted similarity to preceding layers*

   $w_k \leftarrow \alpha^k / \sum_{i=1}^{K} \alpha^i$

   $R^l \leftarrow \sum_{k=1}^{K} w_k \cdot Sim(\boldsymbol{H}^l, \boldsymbol{H}^{l-k})$

   *// Layer importance score*

   $I^l \leftarrow (1 - R^l) \cdot (1 - \bar{s}^l)$

**end for**

**Let** $\mathcal{S}$ be the indices of the Top-85% layers ranked by $Dist^l$.

**for** $l = 0$ **to** $L - 1$ **do**

   **if** $l \notin \mathcal{S}$ **then** $I^l \leftarrow 0$

**end for**

*// Final ranking of critical layers*

**Rank** all layers in descending order according to $I^l$.

---

## C.2. Model Family Configurations

Important architectural parameters are reported :model layers, hidden channels, total learnable parameters, attention heads, positional encoding type, and embedding/prediction layers. All models use patch size 16 and stride 8 for time series embedding with linear input/output projections. We employ the commonly used setting of 336 input steps and 96 steps for future prediction.

*Table 11.* Model Family Configurations. Model configurations across two families, LLM4TS and TSFMs, varying in scale from Tiny to Large. "–" indicates no pre-training (trained from scratch).

| Scale | Model family | Layers | Backbone | Channels | Learnable parameters | Patch Size | Stride |
|-------|-------------|--------|----------|----------|---------------------|------------|--------|
| Tiny | LLM4TS / TSFMs | 6 / 6 | GPT-2 / - | 768 / 768 | 3.92M / 85M | 16 | 8 |
| Small | LLM4TS / TSFMs | 12 / 12 | GPT-2 / - | 768 / 768 | 3.93M / 128M | 16 | 8 |
| Base | LLM4TS / TSFMs | 28 / 28 | Qwen-3 / - | 1,024 / 1,024 | 0.16B / 0.60B | 16 | 8 |
| Large | LLM4TS / TSFMs | 28 / 28 | Qwen-3 / - | 2,048 / 2,048 | 0.32B / 1.73B | 16 | 8 |

We select two representative LLMs as backbones for the LLM4TS family: GPT-2 (Radford et al., 2018) and Qwen-3 (Yang et al., 2025). GPT-2 is a decoder-only Transformer pretrained on large-scale web text, employing learnable absolute positional embeddings and layer normalization before attention . Its simplicity, widespread adoption, and availability in multiple scales make it an ideal baseline for studying LLM adaptation to TS. Qwen-3 is a state-of-the-art Chinese-English bilingual LLM that adopts RoPE and RMSNorm, demonstrating superior long-context modeling and instruction-following capabilities. We include Qwen-3 to examine whether modern architectural advances, particularly relative position awareness and better scaling, and translate into improved TS forecasting performance. By repurposing LLMs without modifying their core architecture, we establish a strong, reproducible baseline for evaluating how well pretrained linguistic representations can be transferred to temporal forecasting.

### C.3. Implementation Details for Sec. 3.2 & 3.3

In Sec. 3.2, all experiments are implemented using the `HuggingFace Transformers` library (v4.51.3) and trained with `DeepSpeed` (v0.14.0) and `Accelerate` (v0.28.0) for distributed training (Wolf et al., 2020). We use either $4\times$ NVIDIA A100 40GB GPUs or $4\times$ NVIDIA H100 80GB GPUs, depending on model scale. Training is conducted with a fixed random seed across all runs to ensure reproducibility. We employ the AdamW optimizer (Loshchilov & Hutter, 2019) with learning rate $1 \times 10^{-4}$, $\beta_1 = 0.9$, $\beta_2 = 0.95$, and $\epsilon = 1 \times 10^{-6}$. Weight decay is set to 0.01. The learning rate follows a cosine decay schedule without warmup. Training uses `bf16` mixed precision and DeepSpeed ZeRO Stage 2 for memory efficiency, with default gradient clipping. Gradient accumulation steps are set to 1, and micro batch size per GPU is fixed at 128. Early stopping is applied based on MSE of validation sets, with a patience of 3 epochs. All components are built on standard HuggingFace and PyTorch APIs.

In Sec. 3.3, to investigate whether the performance advantages of advanced backbone architectures are contingent on abundant training data, we conduct a controlled data-scaling study. We vary the proportion of training data sampled from each dataset — from 20% to 100%, in increments of 20%, while keeping all other experimental settings identical to Sec. 3.2. Validation and test sets remain fixed and unchanged across runs to ensure consistent evaluation.

### C.4. Implementation Details for Sec. 3.4

In the cross-dataset learning setting, we combine the training partitions of all source datasets into a unified corpus. To avoid bias introduced by dataset ordering or temporal structure during training, we fully shuffle all samples globally before each epoch, ensuring that batches are statistically diverse and model updates are not dominated by any single domain. Due to the large volume of combined TS data and memory constraints, we preprocess and store all datasets in `Apache Arrow` format using the `datasets` library (v4.0.0), which enables efficient, zero-copy data loading and minimizes I/O bottlenecks.

All models are trained on $4\times$ NVIDIA H200 140GB GPUs, using the same codebase and hyperparameters as in Sec. 3.2, with the following specific adjustments: per-GPU batch size is set to 128, training is capped at a maximum of 3 epochs, and early stopping is triggered if no improvement in validation MSE is observed for 1 epoch. No curriculum learning, dataset weighting, or domain balancing strategy is applied. Models learn from uniformly sampled batches across all source domains, testing their raw capacity to acquire generalizable temporal representations under maximal data diversity.

### C.5. Implementation Details for Sec. 5.1

For four LLM4TS baselines in Sec. 5, we adopt channel-independent strategy, each variable is processed as an independent sequence. All reproduction and pruning experiments strictly adhere to same strategy and training hyperparameters reported in original papers, ensuring faithful re-implementation. Critically, the selection of trainable versus frozen modules is held identical across all models and experimental phases. For four TSFMs baselines, All models are fine-tuned on the downstream datasets to improve their generalization capability and to align the parameter distributions before and after pruning, thereby ensuring a fair and meaningful comparison of pruning effects. This eliminates confounding factors arising from training protocol differences and allows us to attribute performance changes solely to architectural modification.

### C.6. Hyperparameter Analysis

**Decay factor.** The decay factor $\alpha$ in $R^l$ controls how similarities with preceding layers are aggregated, reflecting the intuition that redundancy should be measured more strongly against nearby representations. While an exponential decay provides a natural and minimal inductive bias, it is important to verify that our conclusions do not depend on a particular

choice of $\alpha$. To this end, we conduct a robustness analysis by varying $\alpha$ over a wide range and examining the stability of the layer importance scores $I^l$. Specifically, we use $\alpha = 0.5$ as a reference, recompute $I^l$ for different values of $\alpha$, and measure the consistency of the resulting layer rankings using Spearman rank correlation, as shown in Fig. 12. The resulting curve shows that the rankings remain highly correlated across a broad range of decay factors, indicating that our analysis is robust to the choice of $\alpha$ and primarily depends on the relative ordering of layers rather than the exact weighting scheme.

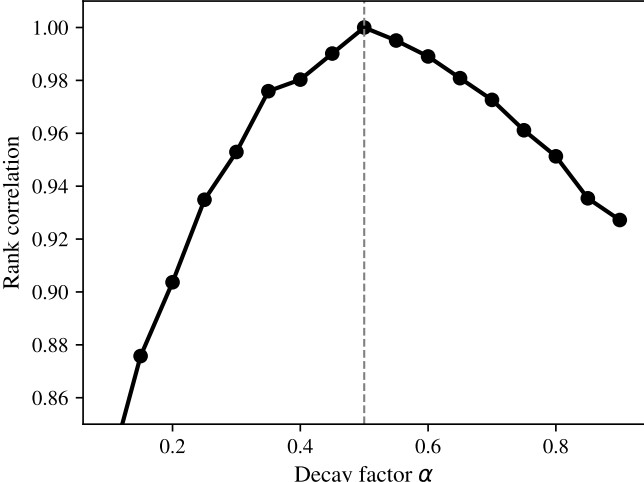

*Figure 12.* Stability of layer importance ranking under different decay factors $\alpha$. The y-axis shows the Spearman rank correlation between the layer rankings induced by $I^l$ at a given $\alpha$ and the reference ranking at $\alpha = 0.5$.

Fig. 13 shows how the decay factor $\alpha$ controls the effective temporal receptive field of the redundancy measure. Setting $\alpha = 0$ assigns all weight to the immediately preceding layer, making it optimal for capturing purely local novelty. However, this choice ignores redundancy with more distant layers and thus fails to capture global redundancy across depth. Non-zero values of $\alpha$ provide a principled trade-off by emphasizing nearby layers while retaining limited historical context.

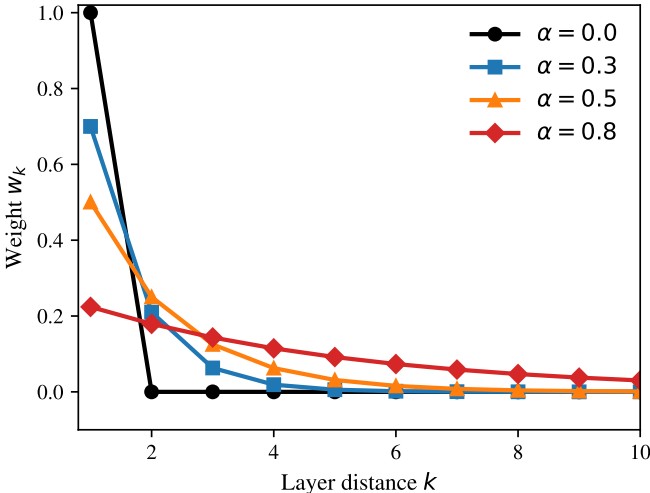

*Figure 13.* Effect of the decay factor $\alpha$ on the weighting of similarities across preceding layers. Larger $\alpha$ assigns non-negligible weights to more distant layers, while $\alpha = 0$ reduces the measure to a purely local comparison with the immediately preceding layer.

# D. Baselines

## D.1. Baselines of LLM4TS

FSCA(Hu et al., 2025): Introducing Context-Alignment to comprehend TS with the same structural and logical awareness they apply to natural language. It achieves this through two complementary alignment mechanisms: (1) *Structural Alignment*, which employs dual-scale graph nodes to capture the hierarchical composition of TS, allowing LLMs to treat long sequences as coherent linguistic units while preserving fine-grained temporal features; and (2) *Logical Alignment*, which utilizes directed edges to model semantic dependencies across variables or time steps, ensuring contextual coherence and relational consistency.

CALF(Liu et al., 2025b): A dual-branch framework is employed, where aligned textual tokens and projected TS tokens are processed through the same frozen pretrained LLM layers to extract semantically aligned features. Cross-modal interaction is established via three alignment mechanisms: linear projection, cross-attention fusion, and contrastive tuning, enabling temporal forecasting grounded in linguistic priors without modifying the original LLM parameters.

TIME-LLM(Jin et al., 2024): The pretrained LLM is kept entirely frozen, and TS forecasting is enabled through two lightweight trainable modules, *Patch Reprogramming* for input adaptation and *Output Projection* for prediction mapping. It adopt a channel-independent strategy that decomposes multivariate forecasting into parallel univariate tasks.

OFA(Zhou et al., 2023): A unified TS analysis framework is established by repurposing a frozen pre-trained language model without modifying its internal Transformer layers. By treating diverse TS tasks, including short- and long-term forecasting, classification, imputation, and anomaly detection, as sequence modeling problems within a common interface, the approach achieves highly competitive performance across all settings. Input TS are projected into the embedding space via lightweight trainable adapters, while outputs are mapped back through task-specific heads, preserving the LLM's original parameters throughout.

## D.2. Baselines of TSFMs

SUNDIAL(Liu et al., 2025c): A family of native TS foundation models is pre-trained on TimeBench, a corpus containing one trillion time points from real-world and synthetic datasets. Training employs a flow-matching-based TimeFlow Loss that enables direct optimization on continuous-valued sequences without discrete tokenization. By modeling the next-patch distribution in a non-parametric generative manner, the approach supports multi-modal probabilistic forecasting and effectively mitigates mode collapse. With only minimal adaptations to the Transformer architecture, the models accept arbitrary-length inputs, achieve strong scalability, and deliver state-of-the-art performance in both point and probabilistic forecasting, while enabling zero-shot inference in just a few milliseconds.

CHRONOS(Ansari et al., 2024): It adapts TS to Transformer-based models through a two-step preprocessing pipeline of scaling and quantization. First, mean-scaling normalizes each value by the mean absolute magnitude of its historical context, ensuring comparability across series. Second, quantization discretizes scaled values into bins, yielding a sequence of discrete tokens. These tokens are modeled using a T5-based architecture trained with cross-entropy loss, enabling the acquisition of general-purpose TS representations while fully leveraging modeling paradigm.

MOIRAI(Woo et al., 2024): It is a large-scale universal time series forecasting model designed to overcome the limitations of the traditional one-model-per-dataset paradigm. By introducing architectural innovations to the standard Transformer, MOIRAI effectively handles cross-frequency learning, supports arbitrary multivariate input dimensions, and adapts to diverse distributional patterns across large-scale datasets. Trained on the Large-scale Open Time Series Archive containing over 27 billion observations across nine domains, MOIRAI demonstrates strong zero-shot forecasting capability, achieving performance comparable to or even surpassing fully fine-tuned models.

TIMES-FM(Das et al., 2024): As a patch-based forecasting framework, it supports variable context lengths and enables output patches longer than input patches. During training, model learns to predict extended horizons by conditioning on progressively longer prefixes. At inference time, long-term forecasting is performed in a semi-autoregressive manner: given a 256-step context, it predicts next 128 steps, then appends predictions to original context to forecast subsequent 128 steps. Internally, each decoding step processes current input patch through an input residual block, adds a positional encoding vector, feeds the result into a stacked Transformer with causal self-attention to ensure temporal consistency, and finally passes the representation through an output residual block to produce the output patch, which is compared against ground truth for loss computation.

We report the detailed performance of the eight baseline models in Table 12. For a fair comparison, we follow the original settings and input–output formats of all baseline models throughout our experiments.

*Table 12.* Based on these models, we perform layer identification and pruning across a total of 13 data distributions.

| | Model | Backbone | Paradigm | Token Level | Channels | Prompt | Alignment | Input Length | Horizons |
|---|---|---|---|---|---|---|---|---|---|
| **LLM4TS** | FSCA | GPT-2 | decoder-only | Patch | 768 | ✔ | sentence | 512 | $\{96, 192, 336, 720\}$ |
| | CALF | GPT-2 | decoder-only | Variable | 768 | ✘ | word | 96 | $\{96, 192, 336, 720\}$ |
| | Time-LLM(G) | GPT-2 | decoder-only | Patch | 768 | ✔ | word | 512 | $\{96, 192, 336, 720\}$ |
| | OFA | GPT-2 | decoder-only | Patch | 768 | ✘ | latent space | 336 | $\{96, 192, 336, 720\}$ |

| | Model | Backbone | Paradigm | Token Level | Channels | Pre-training Scale | Model Size | Input Length | Horizons |
|---|---|---|---|---|---|---|---|---|---|
| **TSFMs** | Sundial$_{Large}$ | $Layers : 24$ | decoder-only | Patch | 1,024 | 1,000B | 450M | 336 | $\{96, 192, 336, 720\}$ |
| | Chronos$_{Base}$ | $Layers : 24$ | encoder-decoder | Point | 1,024 | 200B | 200M | 96 | $\{96, 192, 336, 720\}$ |
| | MOIRAI$_{Large}$ | $Layers : 24$ | encoder-only | Patch | 1,024 | 230B | 300M | 96 | $\{96, 192, 336, 720\}$ |
| | TimesFM | $Layers : 50$ | decoder-only | Patch | 1,280 | 100B | 500M | 96 | $\{96, 192, 336, 720\}$ |

# E. Full Results

## E.1. Full Results for Sec. 3.2

*Table 13.* Full results of performance of LLM4TS and TSFM families across different backbones.

| Models | | MAE | | | | | | | | | MSE | | | | | | | | |
|---|---|---|---|---|---|---|---|---|---|---|---|---|---|---|---|---|---|---|---|
| Family | Scales | ETTh1 | ETTh2 | ETTm1 | ETTm2 | ECL | Exchange | Solar | Weather | avg | ETTh1 | ETTh2 | ETTm1 | ETTm2 | ECL | Exchange | Solar | Weather | avg |
| **LLM4TS** | Tiny | 0.398 | 0.348 | 0.345 | 0.260 | 0.239 | 0.211 | 0.245 | 0.205 | **0.281** | 0.378 | 0.290 | 0.291 | 0.169 | 0.139 | 0.089 | 0.197 | 0.156 | **0.214** |
| | Small | 0.421 | 0.364 | 0.354 | 0.264 | 0.237 | 0.210 | 0.250 | 0.201 | **0.288** | 0.420 | 0.307 | 0.299 | 0.179 | 0.139 | 0.089 | 0.197 | 0.151 | **0.223** |
| | Base | 0.525 | 0.423 | 0.401 | 0.284 | 0.230 | 0.276 | 0.257 | 0.205 | **0.325** | 0.594 | 0.390 | 0.393 | 0.214 | 0.135 | 0.144 | 0.202 | 0.156 | **0.278** |
| | Large | 0.527 | 0.425 | 0.404 | 0.289 | 0.226 | 0.287 | 0.260 | 0.208 | **0.328** | 0.595 | 0.405 | 0.395 | 0.223 | 0.131 | 0.160 | 0.209 | 0.160 | **0.285** |
| **TSFMs** | Tiny | 0.430 | 0.360 | 0.355 | 0.264 | 0.227 | 0.223 | 0.235 | 0.206 | **0.288** | 0.420 | 0.310 | 0.304 | 0.177 | 0.133 | 0.096 | 0.188 | 0.153 | **0.223** |
| | Small | 0.450 | 0.364 | 0.358 | 0.265 | 0.228 | 0.230 | 0.235 | 0.207 | **0.292** | 0.464 | 0.307 | 0.310 | 0.184 | 0.133 | 0.094 | 0.183 | 0.154 | **0.229** |
| | Base | 0.469 | 0.384 | 0.372 | 0.268 | 0.228 | 0.238 | 0.252 | 0.211 | **0.303** | 0.482 | 0.316 | 0.334 | 0.185 | 0.135 | 0.133 | 0.239 | 0.162 | **0.248** |
| | Large | 0.478 | 0.388 | 0.375 | 0.271 | 0.234 | 0.253 | 0.252 | 0.212 | **0.308** | 0.506 | 0.325 | 0.333 | 0.183 | 0.140 | 0.141 | 0.242 | 0.164 | **0.254** |

Result shows enlarging the scale does not confer any performance advantage within each family, but results in performance degradation, shown in Table 13.

## E.2. Full Results for Sec. 3.3

When the volume of training data increases, both MAE and MSE show a consistent downward trend, indicating that the models are able to learn more robust and generalizable representations. Interestingly, despite having a more advanced backbone, Larger-scale models do not always outperform their weaker counterparts under the same data regimes, shown in Table 14 & 15.

## E.3. Full Results for Sec. 3.4

Full results of single-dataset learning are shown in Appendix E.1. Full results of in-domain performance in cross-dataset learning are shown in Table 16, and full results of out-of-domain performance in cross-dataset learning are shown in Table 17.

While we might expect that exposure to a wider variety of TS patterns would enhance models' generalization ability, results suggest that both families are largely limited by other factors. Increasing the diversity of training data does not necessarily lead to corresponding performance improvements, neither on in-domain nor out-of-domain tasks, shown in Table 16 & 17.

*Table 14.* Performance of LLM4TS families at Small and Base scales under varying training data ratios.

| Models | Ratio | MAE | | | | | | | | | MSE | | | | | | | | |
|---|---|---|---|---|---|---|---|---|---|---|---|---|---|---|---|---|---|---|---|
| | | ETTh1 | ETTh2 | ETTm1 | ETTm2 | ECL | Exchange | Solar | Weather | avg | ETTh1 | ETTh2 | ETTm1 | ETTm2 | ECL | Exchange | Solar | Weather | avg |
| LLM4TS-Small | 0.2 | 0.425 | 0.386 | 0.351 | 0.268 | 0.244 | 0.234 | 0.257 | 0.215 | **0.297** | 0.418 | 0.348 | 0.298 | 0.183 | 0.144 | 0.108 | 0.197 | 0.162 | **0.232** |
| | 0.4 | 0.439 | 0.365 | 0.356 | 0.267 | 0.234 | 0.217 | 0.247 | 0.218 | **0.293** | 0.445 | 0.320 | 0.305 | 0.179 | 0.139 | 0.092 | 0.195 | 0.168 | **0.230** |
| | 0.6 | 0.421 | 0.351 | 0.362 | 0.276 | 0.237 | 0.225 | 0.251 | 0.212 | **0.292** | 0.413 | 0.298 | 0.313 | 0.188 | 0.139 | 0.099 | 0.213 | 0.158 | **0.227** |
| | 0.8 | 0.419 | 0.360 | 0.361 | 0.261 | 0.237 | 0.221 | 0.253 | 0.204 | **0.290** | 0.411 | 0.307 | 0.308 | 0.174 | 0.139 | 0.095 | 0.194 | 0.153 | **0.222** |
| | 1.0 | 0.421 | 0.364 | 0.354 | 0.264 | 0.237 | 0.210 | 0.250 | 0.201 | **0.288** | 0.420 | 0.307 | 0.299 | 0.179 | 0.139 | 0.089 | 0.197 | 0.151 | **0.223** |
| LLM4TS-Base | 0.2 | 0.609 | 0.500 | 0.450 | 0.317 | 0.258 | 0.289 | 0.268 | 0.266 | **0.370** | 0.756 | 0.529 | 0.443 | 0.238 | 0.159 | 0.163 | 0.197 | 0.228 | **0.339** |
| | 0.4 | 0.595 | 0.459 | 0.447 | 0.334 | 0.241 | 0.269 | 0.282 | 0.266 | **0.362** | 0.742 | 0.450 | 0.444 | 0.254 | 0.144 | 0.140 | 0.199 | 0.225 | **0.325** |
| | 0.6 | 0.542 | 0.434 | 0.419 | 0.301 | 0.235 | 0.278 | 0.264 | 0.249 | **0.340** | 0.617 | 0.406 | 0.398 | 0.217 | 0.139 | 0.143 | 0.210 | 0.204 | **0.292** |
| | 0.8 | 0.533 | 0.433 | 0.420 | 0.304 | 0.232 | 0.272 | 0.270 | 0.227 | **0.336** | 0.600 | 0.415 | 0.413 | 0.215 | 0.138 | 0.138 | 0.207 | 0.179 | **0.288** |
| | 1.0 | 0.525 | 0.423 | 0.401 | 0.284 | 0.230 | 0.276 | 0.257 | 0.205 | **0.325** | 0.594 | 0.390 | 0.393 | 0.214 | 0.135 | 0.144 | 0.202 | 0.156 | **0.278** |

*Table 15.* Performance of TSFMs families at Tiny and Large scales under varying training data ratios.

| Models | Ratio | MAE | | | | | | | | | MSE | | | | | | | | |
|---|---|---|---|---|---|---|---|---|---|---|---|---|---|---|---|---|---|---|---|
| | | ETTh1 | ETTh2 | ETTm1 | ETTm2 | ECL | Exchange | Solar | Weather | avg | ETTh1 | ETTh2 | ETTm1 | ETTm2 | ECL | Exchange | Solar | Weather | avg |
| TSFMs-Tiny | 0.2 | 0.464 | 0.466 | 0.372 | 0.290 | 0.237 | 0.270 | 0.260 | 0.254 | **0.327** | 0.480 | 0.481 | 0.320 | 0.199 | 0.141 | 0.137 | 0.212 | 0.202 | **0.271** |
| | 0.4 | 0.446 | 0.388 | 0.360 | 0.284 | 0.232 | 0.236 | 0.255 | 0.221 | **0.302** | 0.453 | 0.337 | 0.308 | 0.193 | 0.137 | 0.113 | 0.207 | 0.171 | **0.240** |
| | 0.6 | 0.446 | 0.387 | 0.361 | 0.281 | 0.229 | 0.229 | 0.258 | 0.232 | **0.303** | 0.461 | 0.339 | 0.307 | 0.189 | 0.135 | 0.103 | 0.202 | 0.179 | **0.239** |
| | 0.8 | 0.450 | 0.370 | 0.357 | 0.286 | 0.228 | 0.238 | 0.239 | 0.218 | **0.298** | 0.461 | 0.325 | 0.305 | 0.192 | 0.134 | 0.108 | 0.204 | 0.167 | **0.237** |
| | 1.0 | 0.430 | 0.360 | 0.355 | 0.264 | 0.227 | 0.223 | 0.235 | 0.206 | **0.287** | 0.420 | 0.310 | 0.304 | 0.177 | 0.133 | 0.096 | 0.188 | 0.153 | **0.223** |
| TSFMs-Large | 0.2 | 0.499 | 0.417 | 0.409 | 0.290 | 0.244 | 0.311 | 0.273 | 0.238 | **0.335** | 0.524 | 0.385 | 0.380 | 0.208 | 0.149 | 0.188 | 0.252 | 0.184 | **0.284** |
| | 0.4 | 0.532 | 0.401 | 0.387 | 0.287 | 0.238 | 0.246 | 0.260 | 0.222 | **0.321** | 0.610 | 0.351 | 0.352 | 0.200 | 0.145 | 0.112 | 0.243 | 0.168 | **0.273** |
| | 0.6 | 0.478 | 0.373 | 0.364 | 0.273 | 0.233 | 0.242 | 0.256 | 0.230 | **0.306** | 0.499 | 0.326 | 0.317 | 0.186 | 0.139 | 0.113 | 0.240 | 0.184 | **0.250** |
| | 0.8 | 0.450 | 0.370 | 0.365 | 0.272 | 0.232 | 0.266 | 0.245 | 0.219 | **0.302** | 0.465 | 0.323 | 0.337 | 0.188 | 0.139 | 0.136 | 0.231 | 0.168 | **0.248** |
| | 1.0 | 0.478 | 0.388 | 0.375 | 0.271 | 0.234 | 0.253 | 0.252 | 0.212 | **0.308** | 0.506 | 0.325 | 0.333 | 0.183 | 0.140 | 0.141 | 0.242 | 0.164 | **0.254** |

In some cases, even with more diverse data, models exhibit only marginal gains or plateau in performance, meaning simply expanding data diversity is insufficient to fully exploit the potential of backbones.

### E.4. Full Results for Sec. 4.2

In Appendix E.1, we report in-domain performance of models under single-dataset learning. In Appendix E.3, we present in-domain and out-of-domain performance under cross-dataset learning. In this subsection, we report the full results obtained after extracting only a few Transformer layers and re-aligning them on the original training sets.

Retaining only the critical Transformer layers, followed by fine-tuning on original training sets, not only preserves but in some cases even improves forecasting accuracy. Our approach effectively filters out less informative representations, allowing model to focus on the most salient temporal dependencies. In addition to maintaining predictive performance, it significantly reduces the total number of parameters, which in turn lowers memory footprint and accelerates inference. Full results are reported in Table 18- 20.

### E.5. Full Results for Sec. 5.1

We sequentially retain critical layers of four LLM4TS models—FSCA, CALF, TIME-LLM(G), and OFA, and perform re-alignment on the original training sets. In the following, we provide full results over horizons $\in \{96, 192, 336, 720\}$. We do not modify training strategies for the LLM4TS , retaining original experiment settings for either frozen or trainable parameters. For example, in the case of FSCA, original implementation fine-tuned the Layer-Norm and Word Positional Encoding, freezing MHSA and FFN to avoid catastrophic forgetting. We maintained this strategy for our experiments. Conversely, for TIME-LLM, the entire model was frozen during training, and we followed this same approach in our experiments. Full results are reported in Table 21- 24.

*Table 16.* Full results of in-domain performance in cross-dataset learning.

| Models | | | | | | MAE | | | | | | | | | | MSE | | | | |
|---|---|---|---|---|---|---|---|---|---|---|---|---|---|---|---|---|---|---|---|---|
| Family | Scales | ETTh1 | ETTh2 | ETTm1 | ETTm2 | ECL | Exchange | Solar | Weather | avg | ETTh1 | ETTh2 | ETTm1 | ETTm2 | ECL | Exchange | Solar | Weather | avg |
| LLM4TS | Small-C | 0.405 | 0.355 | 0.400 | 0.275 | 0.244 | 0.229 | 0.268 | 0.212 | **0.299** | 0.395 | 0.307 | 0.372 | 0.191 | 0.146 | 0.106 | 0.207 | 0.169 | **0.237** |
| LLM4TS | Base-C | 0.401 | 0.347 | 0.380 | 0.270 | 0.233 | 0.223 | 0.251 | 0.211 | **0.289** | 0.386 | 0.294 | 0.345 | 0.187 | 0.137 | 0.102 | 0.196 | 0.164 | **0.226** |
| LLM4TS | Large-C | 0.402 | 0.343 | 0.382 | 0.263 | 0.231 | 0.224 | 0.249 | 0.207 | **0.288** | 0.386 | 0.288 | 0.331 | 0.179 | 0.136 | 0.103 | 0.204 | 0.161 | **0.223** |
| TSFMs | Small-C | 0.401 | 0.350 | 0.365 | 0.265 | 0.228 | 0.225 | 0.256 | 0.209 | **0.287** | 0.389 | 0.305 | 0.320 | 0.182 | 0.134 | 0.101 | 0.214 | 0.161 | **0.226** |
| TSFMs | Base-C | 0.435 | 0.358 | 0.377 | 0.275 | 0.232 | 0.239 | 0.256 | 0.218 | **0.299** | 0.447 | 0.313 | 0.352 | 0.195 | 0.140 | 0.110 | 0.265 | 0.175 | **0.250** |
| TSFMs | Large-C | 0.421 | 0.370 | 0.379 | 0.278 | 0.234 | 0.240 | 0.261 | 0.221 | **0.301** | 0.424 | 0.338 | 0.355 | 0.201 | 0.142 | 0.114 | 0.284 | 0.177 | **0.254** |

*Table 17.* Full results of out-of-domain performance in cross-dataset learning.

| Models | | | | | | MAE | | | | | | | | | | MSE | | | | |
|---|---|---|---|---|---|---|---|---|---|---|---|---|---|---|---|---|---|---|---|---|
| Family | Scales | NN5 | PDB | Sceaux | Smart | Spanish | Sunspot Rain | US Births | Wind Power | avg | NN5 | PDB | Sceaux | Smart | Spanish | Sunspot Rain | US Births | Wind Power | avg |
| LLM4TS | Small-C | 0.775 | 0.330 | 0.570 | 0.513 | 0.396 | 0.916 | 0.621 | 0.874 | **0.624** | 1.011 | 0.206 | 0.655 | 0.697 | 0.324 | 1.307 | 0.572 | 1.198 | **0.746** |
| LLM4TS | Base-C | 0.785 | 0.324 | 0.575 | 0.513 | 0.385 | 0.934 | 0.647 | 0.853 | **0.627** | 1.033 | 0.202 | 0.646 | 0.681 | 0.309 | 1.377 | 0.609 | 1.130 | **0.748** |
| LLM4TS | Large-C | 0.791 | 0.320 | 0.574 | 0.513 | 0.384 | 0.932 | 0.661 | 0.852 | **0.628** | 1.051 | 0.197 | 0.646 | 0.686 | 0.305 | 1.341 | 0.636 | 1.131 | **0.749** |
| TSFMs | Small-C | 0.786 | 0.322 | 0.587 | 0.546 | 0.382 | 1.018 | 0.630 | 0.879 | **0.644** | 1.017 | 0.199 | 0.653 | 0.714 | 0.300 | 1.580 | 0.604 | 1.154 | **0.777** |
| TSFMs | Base-C | 0.804 | 0.339 | 0.623 | 0.569 | 0.413 | 0.976 | 0.653 | 0.918 | **0.662** | 1.100 | 0.218 | 0.725 | 0.790 | 0.347 | 1.401 | 0.630 | 1.307 | **0.815** |
| TSFMs | Large-C | 0.827 | 0.350 | 0.628 | 0.575 | 0.412 | 1.037 | 0.683 | 0.942 | **0.682** | 1.156 | 0.236 | 0.741 | 0.821 | 0.345 | 1.607 | 0.671 | 1.402 | **0.872** |

*Table 23.* Full results of TIME-LLM(G). Input length is 512, horizons $\in \{96, 192, 336, 720\}$

| Horizons | 96 | | | | 192 | | | | 336 | | | | 720 | | | | avg | | | |
|---|---|---|---|---|---|---|---|---|---|---|---|---|---|---|---|---|---|---|---|---|
| Metric | Pruned | | Original | | Pruned | | Original | | Pruned | | Original | | Pruned | | Original | | Pruned | | Original | |
| | MAE | MSE | MAE | MSE | MAE | MSE | MAE | MSE | MAE | MSE | MAE | MSE | MAE | MSE | MAE | MSE | MAE | MSE | MAE | MSE |
| **ETTh1** | 0.411 | 0.385 | 0.409 | 0.381 | 0.430 | 0.413 | 0.442 | 0.426 | 0.467 | 0.465 | 0.471 | 0.468 | 0.477 | 0.463 | 0.516 | 0.519 | 0.446 | 0.431 | 0.459 | 0.448 |
| **ETTh2** | 0.355 | 0.291 | 0.357 | 0.298 | 0.387 | 0.354 | 0.404 | 0.368 | 0.428 | 0.393 | 0.426 | 0.394 | 0.463 | 0.433 | 0.452 | 0.419 | 0.408 | 0.368 | 0.410 | 0.370 |
| **ETTm1** | 0.350 | 0.292 | 0.354 | 0.295 | 0.387 | 0.354 | 0.385 | 0.352 | 0.393 | 0.363 | 0.393 | 0.369 | 0.430 | 0.421 | 0.424 | 0.422 | 0.390 | 0.358 | 0.389 | 0.359 |
| **ETTm2** | 0.268 | 0.179 | 0.266 | 0.173 | 0.306 | 0.232 | 0.314 | 0.243 | 0.339 | 0.284 | 0.346 | 0.299 | 0.395 | 0.376 | 0.395 | 0.380 | 0.327 | 0.268 | 0.330 | 0.274 |
| **ECL** | 0.236 | 0.134 | 0.253 | 0.143 | 0.247 | 0.149 | 0.263 | 0.159 | 0.274 | 0.168 | 0.271 | 0.170 | 0.296 | 0.202 | 0.296 | 0.202 | 0.263 | 0.163 | 0.271 | 0.169 |
| **Exchange** | 0.218 | 0.094 | 0.207 | 0.086 | 0.353 | 0.228 | 0.338 | 0.219 | 0.476 | 0.418 | 0.492 | 0.446 | 0.746 | 1.002 | 0.754 | 1.012 | 0.448 | 0.436 | 0.448 | 0.441 |
| **Solar** | 0.254 | 0.179 | 0.256 | 0.179 | 0.248 | 0.183 | 0.247 | 0.183 | 0.256 | 0.193 | 0.255 | 0.189 | 0.268 | 0.206 | 0.270 | 0.205 | 0.257 | 0.190 | 0.257 | 0.189 |
| **Traffic** | 0.262 | 0.361 | 0.268 | 0.365 | 0.273 | 0.382 | 0.273 | 0.381 | 0.283 | 0.399 | 0.290 | 0.402 | 0.299 | 0.436 | 0.307 | 0.442 | 0.279 | 0.395 | 0.284 | 0.398 |
| **Weather** | 0.202 | 0.151 | 0.204 | 0.152 | 0.243 | 0.194 | 0.242 | 0.192 | 0.284 | 0.249 | 0.286 | 0.248 | 0.334 | 0.318 | 0.333 | 0.319 | 0.266 | 0.228 | 0.266 | 0.228 |
| **PEMS03** | 0.248 | 0.143 | 0.248 | 0.143 | 0.269 | 0.169 | 0.268 | 0.171 | 0.288 | 0.185 | 0.291 | 0.184 | 0.324 | 0.237 | 0.329 | 0.242 | 0.282 | 0.183 | 0.284 | 0.185 |
| **PEMS04** | 0.352 | 0.436 | 0.358 | 0.438 | 0.364 | 0.459 | 0.367 | 0.461 | 0.377 | 0.481 | 0.377 | 0.482 | 0.411 | 0.551 | 0.414 | 0.556 | 0.376 | 0.482 | 0.379 | 0.484 |
| **PEMS07** | 0.231 | 0.105 | 0.231 | 0.110 | 0.249 | 0.135 | 0.252 | 0.140 | 0.244 | 0.131 | 0.247 | 0.134 | 0.277 | 0.175 | 0.284 | 0.186 | 0.250 | 0.137 | 0.253 | 0.142 |
| **PEMS08** | 0.335 | 0.446 | 0.362 | 0.472 | 0.382 | 0.527 | 0.381 | 0.529 | 0.384 | 0.536 | 0.395 | 0.566 | 0.406 | 0.598 | 0.408 | 0.610 | 0.377 | 0.527 | 0.386 | 0.544 |

*Table 18.* Full results of performance of LLM4TS and TSFM families across different backbones(After Pruning).

| Models | | MAE | | | | | | | | | MSE | | | | | | | | |
|---|---|---|---|---|---|---|---|---|---|---|---|---|---|---|---|---|---|---|---|
| Family | Scales | ETTh1 | ETTh2 | ETTm1 | ETTm2 | ECL | Exchange | Solar | Weather | avg | ETTh1 | ETTh2 | ETTm1 | ETTm2 | ECL | Exchange | Solar | Weather | avg |
| LLM4TS | Small | 0.404 | 0.362 | 0.352 | 0.264 | 0.231 | 0.210 | 0.245 | 0.199 | **0.283** | 0.388 | 0.315 | 0.297 | 0.174 | 0.135 | 0.088 | 0.197 | 0.151 | **0.218** |
| | Base | 0.437 | 0.365 | 0.354 | 0.265 | 0.224 | 0.247 | 0.250 | 0.200 | **0.293** | 0.419 | 0.310 | 0.300 | 0.179 | 0.131 | 0.115 | 0.204 | 0.150 | **0.226** |
| | Large | 0.412 | 0.365 | 0.350 | 0.262 | 0.223 | 0.207 | 0.248 | 0.200 | **0.283** | 0.402 | 0.315 | 0.296 | 0.168 | 0.129 | 0.092 | 0.198 | 0.152 | **0.219** |
| TSFMs | Small | 0.420 | 0.362 | 0.360 | 0.262 | 0.226 | 0.224 | 0.235 | 0.203 | **0.287** | 0.417 | 0.309 | 0.298 | 0.170 | 0.136 | 0.098 | 0.184 | 0.154 | **0.221** |
| | Base | 0.406 | 0.358 | 0.364 | 0.266 | 0.227 | 0.221 | 0.229 | 0.205 | **0.284** | 0.410 | 0.303 | 0.301 | 0.177 | 0.132 | 0.101 | 0.180 | 0.158 | **0.220** |
| | Large | 0.436 | 0.362 | 0.359 | 0.274 | 0.217 | 0.212 | 0.237 | 0.221 | **0.290** | 0.427 | 0.316 | 0.292 | 0.182 | 0.130 | 0.133 | 0.202 | 0.166 | **0.231** |

*Table 19.* Full results of in-domain performance in cross-dataset learning(After Pruning).

| Models | | MAE | | | | | | | | | MSE | | | | | | | | |
|---|---|---|---|---|---|---|---|---|---|---|---|---|---|---|---|---|---|---|---|
| Family | Scales | ETTh1 | ETTh2 | ETTm1 | ETTm2 | ECL | Exchange | Solar | Weather | avg | ETTh1 | ETTh2 | ETTm1 | ETTm2 | ECL | Exchange | Solar | Weather | avg |
| LLM4TS | Small-C | 0.405 | 0.349 | 0.404 | 0.273 | 0.246 | 0.227 | 0.266 | 0.211 | **0.298** | 0.386 | 0.294 | 0.372 | 0.192 | 0.148 | 0.104 | 0.203 | 0.162 | **0.233** |
| | Base-C | 0.404 | 0.344 | 0.376 | 0.265 | 0.229 | 0.223 | 0.254 | 0.209 | **0.288** | 0.385 | 0.288 | 0.336 | 0.188 | 0.134 | 0.107 | 0.203 | 0.162 | **0.225** |
| | Large-C | 0.399 | 0.347 | 0.380 | 0.262 | 0.230 | 0.220 | 0.246 | 0.204 | **0.286** | 0.383 | 0.290 | 0.336 | 0.173 | 0.132 | 0.132 | 0.197 | 0.166 | **0.226** |
| TSFMs | Small-C | 0.409 | 0.349 | 0.366 | 0.263 | 0.228 | 0.230 | 0.242 | 0.205 | **0.286** | 0.386 | 0.306 | 0.321 | 0.185 | 0.137 | 0.104 | 0.214 | 0.161 | **0.227** |
| | Base-C | 0.415 | 0.354 | 0.371 | 0.274 | 0.234 | 0.225 | 0.254 | 0.217 | **0.293** | 0.426 | 0.308 | 0.354 | 0.190 | 0.141 | 0.109 | 0.273 | 0.175 | **0.247** |
| | Large-C | 0.423 | 0.361 | 0.271 | 0.269 | 0.224 | 0.236 | 0.249 | 0.224 | **0.282** | 0.431 | 0.314 | 0.194 | 0.183 | 0.137 | 0.124 | 0.231 | 0.177 | **0.224** |

*Table 20.* Full results of out-of-domain performance in cross-dataset learning(After Pruning).

| Models | | MAE | | | | | | | | | MSE | | | | | | | | |
|---|---|---|---|---|---|---|---|---|---|---|---|---|---|---|---|---|---|---|---|
| Family | Scales | NN5 | PDB | Sceaux | Smart | Spanish | Sunspot Rain | US Births | Wind Power | avg | NN5 | PDB | Sceaux | Smart | Spanish | Sunspot Rain | US Births | Wind Power | avg |
| LLM4TS | Small-C | 0.769 | 0.327 | 0.573 | 0.513 | 0.391 | 0.946 | 0.616 | 0.865 | **0.625** | 1.010 | 0.201 | 0.657 | 0.688 | 0.316 | 1.321 | 0.574 | 1.156 | **0.740** |
| | Base-C | 0.777 | 0.326 | 0.572 | 0.511 | 0.382 | 0.921 | 0.602 | 0.857 | **0.618** | 1.017 | 0.198 | 0.646 | 0.681 | 0.313 | 1.325 | 0.596 | 1.124 | **0.738** |
| | Large-C | 0.779 | 0.322 | 0.573 | 0.513 | 0.391 | 0.912 | 0.621 | 0.846 | **0.620** | 0.997 | 0.204 | 0.643 | 0.681 | 0.307 | 1.307 | 0.584 | 1.097 | **0.727** |
| TSFMs | Small-C | 0.790 | 0.315 | 0.594 | 0.531 | 0.374 | 0.975 | 0.645 | 0.856 | **0.635** | 1.106 | 0.196 | 0.679 | 0.700 | 0.287 | 1.511 | 0.612 | 1.163 | **0.782** |
| | Base-C | 0.814 | 0.345 | 0.615 | 0.536 | 0.425 | 0.971 | 0.642 | 0.885 | **0.654** | 0.987 | 0.249 | 0.666 | 0.702 | 0.346 | 1.412 | 0.615 | 1.206 | **0.773** |
| | Large-C | 0.830 | 0.337 | 0.622 | 0.541 | 0.385 | 0.984 | 0.629 | 0.946 | **0.659** | 1.143 | 0.222 | 0.722 | 0.724 | 0.298 | 1.456 | 0.605 | 1.396 | **0.821** |

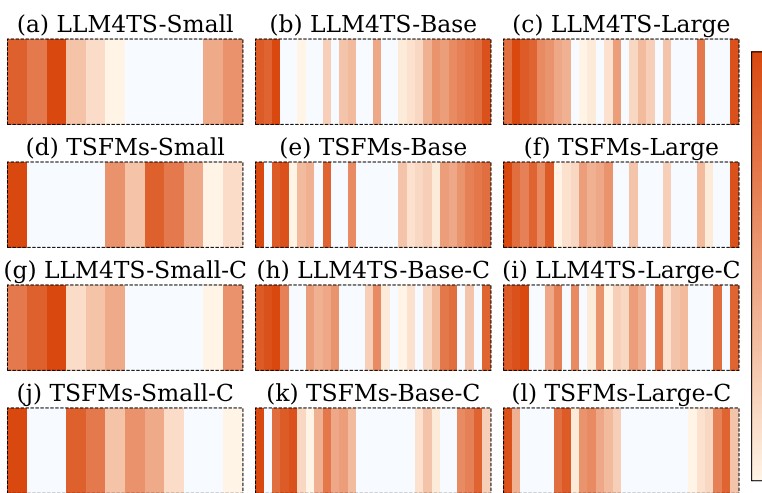

(a) LLM4TS-Small  (b) LLM4TS-Base  (c) LLM4TS-Large
(d) TSFMs-Small  (e) TSFMs-Base  (f) TSFMs-Large
(g) LLM4TS-Small-C  (h) LLM4TS-Base-C  (i) LLM4TS-Large-C
(j) TSFMs-Small-C  (k) TSFMs-Base-C  (l) TSFMs-Large-C

*Figure 14.* Average layer-wise importance, where darker colors indicate greater importance. The key layers are located in the initial and final portions.

*Table 21.* Full results of FSCA. Input length is 512, horizons ∈ {96, 192, 336, 720}

| Horizons | 96 | | | | 192 | | | | 336 | | | | 720 | | | | avg | | | |
|---|---|---|---|---|---|---|---|---|---|---|---|---|---|---|---|---|---|---|---|---|
| Metric | Pruned | | Original | | Pruned | | Original | | Pruned | | Original | | Pruned | | Original | | Pruned | | Original | |
| | MAE | MSE | MAE | MSE | MAE | MSE | MAE | MSE | MAE | MSE | MAE | MSE | MAE | MSE | MAE | MSE | MAE | MSE | MAE | MSE |
| **ETTh1** | 0.404 | 0.372 | 0.397 | 0.365 | 0.436 | 0.415 | 0.433 | 0.418 | 0.447 | 0.434 | 0.451 | 0.442 | 0.484 | 0.484 | 0.496 | 0.494 | 0.443 | 0.426 | 0.444 | 0.430 |
| **ETTh2** | 0.342 | 0.277 | 0.346 | 0.284 | 0.384 | 0.348 | 0.380 | 0.340 | 0.403 | 0.367 | 0.402 | 0.367 | 0.432 | 0.403 | 0.433 | 0.401 | 0.390 | 0.349 | 0.390 | 0.348 |
| **ETTm1** | 0.348 | 0.289 | 0.349 | 0.291 | 0.376 | 0.333 | 0.377 | 0.334 | 0.394 | 0.362 | 0.396 | 0.365 | 0.421 | 0.418 | 0.425 | 0.419 | 0.385 | 0.350 | 0.387 | 0.352 |
| **ETTm2** | 0.256 | 0.168 | 0.262 | 0.176 | 0.298 | 0.227 | 0.297 | 0.222 | 0.336 | 0.280 | 0.339 | 0.281 | 0.384 | 0.357 | 0.385 | 0.357 | 0.319 | 0.258 | 0.320 | 0.259 |
| **ECL** | 0.240 | 0.138 | 0.237 | 0.134 | 0.251 | 0.151 | 0.252 | 0.152 | 0.268 | 0.164 | 0.271 | 0.169 | 0.306 | 0.208 | 0.306 | 0.209 | 0.266 | 0.165 | 0.267 | 0.166 |
| **Exchange** | 0.202 | 0.087 | 0.213 | 0.090 | 0.329 | 0.205 | 0.342 | 0.222 | 0.453 | 0.381 | 0.462 | 0.396 | 0.751 | 1.070 | 0.779 | 1.092 | 0.434 | 0.436 | 0.449 | 0.450 |
| **Solar** | 0.255 | 0.192 | 0.260 | 0.195 | 0.254 | 0.190 | 0.266 | 0.204 | 0.270 | 0.201 | 0.284 | 0.222 | 0.277 | 0.210 | 0.286 | 0.220 | 0.264 | 0.198 | 0.274 | 0.210 |
| **Traffic** | 0.258 | 0.355 | 0.273 | 0.369 | 0.269 | 0.377 | 0.274 | 0.382 | 0.279 | 0.391 | 0.283 | 0.397 | 0.302 | 0.438 | 0.308 | 0.440 | 0.277 | 0.390 | 0.284 | 0.397 |
| **Weather** | 0.204 | 0.150 | 0.205 | 0.151 | 0.243 | 0.195 | 0.246 | 0.195 | 0.297 | 0.267 | 0.285 | 0.249 | 0.331 | 0.319 | 0.338 | 0.322 | 0.269 | 0.233 | 0.268 | 0.229 |
| **PEMS03** | 0.241 | 0.137 | 0.246 | 0.138 | 0.255 | 0.153 | 0.258 | 0.158 | 0.265 | 0.179 | 0.268 | 0.171 | 0.295 | 0.216 | 0.306 | 0.218 | 0.264 | 0.171 | 0.270 | 0.171 |
| **PEMS04** | 0.315 | 0.405 | 0.330 | 0.409 | 0.334 | 0.436 | 0.346 | 0.441 | 0.354 | 0.465 | 0.360 | 0.468 | 0.383 | 0.525 | 0.395 | 0.536 | 0.346 | 0.458 | 0.358 | 0.463 |
| **PEMS07** | 0.197 | 0.089 | 0.223 | 0.104 | 0.213 | 0.104 | 0.233 | 0.121 | 0.227 | 0.119 | 0.245 | 0.133 | 0.246 | 0.147 | 0.269 | 0.163 | 0.221 | 0.115 | 0.242 | 0.130 |
| **PEMS08** | 0.316 | 0.438 | 0.326 | 0.434 | 0.335 | 0.485 | 0.340 | 0.488 | 0.352 | 0.534 | 0.364 | 0.528 | 0.376 | 0.584 | 0.383 | 0.586 | 0.345 | 0.510 | 0.353 | 0.509 |

*Table 22.* Full results of CALF. Input length is 96, horizons ∈ {96, 192, 336, 720}

| Horizons | 96 | | | | 192 | | | | 336 | | | | 720 | | | | avg | | | |
|---|---|---|---|---|---|---|---|---|---|---|---|---|---|---|---|---|---|---|---|---|
| Metric | Pruned | | Original | | Pruned | | Original | | Pruned | | Original | | Pruned | | Original | | Pruned | | Original | |
| | MAE | MSE | MAE | MSE | MAE | MSE | MAE | MSE | MAE | MSE | MAE | MSE | MAE | MSE | MAE | MSE | MAE | MSE | MAE | MSE |
| **ETTh1** | 0.391 | 0.376 | 0.394 | 0.380 | 0.422 | 0.413 | 0.421 | 0.427 | 0.440 | 0.471 | 0.443 | 0.473 | 0.472 | 0.483 | 0.477 | 0.505 | 0.431 | 0.436 | 0.434 | 0.446 |
| **ETTh2** | 0.331 | 0.286 | 0.335 | 0.290 | 0.381 | 0.357 | 0.384 | 0.367 | 0.419 | 0.397 | 0.423 | 0.414 | 0.432 | 0.408 | 0.437 | 0.421 | 0.391 | 0.362 | 0.395 | 0.373 |
| **ETTm1** | 0.340 | 0.315 | 0.344 | 0.317 | 0.371 | 0.368 | 0.375 | 0.372 | 0.391 | 0.396 | 0.394 | 0.400 | 0.432 | 0.468 | 0.436 | 0.474 | 0.384 | 0.387 | 0.387 | 0.391 |
| **ETTm2** | 0.253 | 0.166 | 0.251 | 0.174 | 0.294 | 0.229 | 0.295 | 0.239 | 0.338 | 0.295 | 0.333 | 0.297 | 0.395 | 0.381 | 0.392 | 0.398 | 0.320 | 0.268 | 0.318 | 0.277 |
| **ECL** | 0.245 | 0.165 | 0.248 | 0.165 | 0.258 | 0.173 | 0.258 | 0.174 | 0.270 | 0.186 | 0.274 | 0.190 | 0.303 | 0.212 | 0.307 | 0.228 | 0.269 | 0.184 | 0.272 | 0.189 |
| **Exchange** | 0.201 | 0.084 | 0.201 | 0.084 | 0.299 | 0.178 | 0.298 | 0.177 | 0.437 | 0.361 | 0.424 | 0.344 | 0.714 | 0.891 | 0.708 | 0.877 | 0.413 | 0.378 | 0.408 | 0.370 |
| **Solar** | 0.245 | 0.205 | 0.248 | 0.211 | 0.263 | 0.244 | 0.267 | 0.267 | 0.259 | 0.233 | 0.280 | 0.262 | 0.273 | 0.226 | 0.279 | 0.262 | 0.260 | 0.227 | 0.268 | 0.251 |
| **Traffic** | 0.272 | 0.425 | 0.275 | 0.439 | 0.274 | 0.401 | 0.279 | 0.452 | 0.284 | 0.416 | 0.285 | 0.467 | 0.305 | 0.455 | 0.304 | 0.501 | 0.284 | 0.424 | 0.286 | 0.465 |
| **Weather** | 0.210 | 0.174 | 0.215 | 0.179 | 0.252 | 0.220 | 0.254 | 0.223 | 0.285 | 0.273 | 0.295 | 0.279 | 0.336 | 0.353 | 0.345 | 0.355 | 0.271 | 0.255 | 0.277 | 0.259 |
| **PEMS03** | 0.393 | 0.341 | 0.395 | 0.348 | 0.419 | 0.394 | 0.423 | 0.398 | 0.378 | 0.335 | 0.380 | 0.340 | 0.412 | 0.383 | 0.427 | 0.407 | 0.401 | 0.363 | 0.406 | 0.373 |
| **PEMS04** | 0.505 | 0.763 | 0.510 | 0.766 | 0.529 | 0.810 | 0.534 | 0.815 | 0.474 | 0.709 | 0.479 | 0.714 | 0.523 | 0.807 | 0.530 | 0.816 | 0.508 | 0.772 | 0.513 | 0.778 |
| **PEMS07** | 0.401 | 0.375 | 0.404 | 0.388 | 0.416 | 0.427 | 0.430 | 0.432 | 0.365 | 0.333 | 0.370 | 0.341 | 0.396 | 0.403 | 0.418 | 0.411 | 0.394 | 0.384 | 0.405 | 0.393 |
| **PEMS08** | 0.479 | 0.760 | 0.491 | 0.777 | 0.518 | 0.856 | 0.522 | 0.862 | 0.463 | 0.762 | 0.485 | 0.811 | 0.493 | 0.702 | 0.525 | 0.899 | 0.488 | 0.770 | 0.506 | 0.837 |

*Table 24.* Full results of OFA. Input length is 336, horizons $\in \{96, 192, 336, 720\}$

| Horizons | 96 | | | | 192 | | | | 336 | | | | 720 | | | | avg | | | |
|---|---|---|---|---|---|---|---|---|---|---|---|---|---|---|---|---|---|---|---|---|
| Metric | Pruned | | Original | | Pruned | | Original | | Pruned | | Original | | Pruned | | Original | | Pruned | | Original | |
| | MAE | MSE | MAE | MSE | MAE | MSE | MAE | MSE | MAE | MSE | MAE | MSE | MAE | MSE | MAE | MSE | MAE | MSE | MAE | MSE |
| **ETTh1** | 0.401 | 0.382 | 0.398 | 0.378 | 0.424 | 0.422 | 0.420 | 0.417 | 0.448 | 0.458 | 0.436 | 0.444 | 0.465 | 0.455 | 0.481 | 0.482 | 0.435 | 0.429 | 0.434 | 0.430 |
| **ETTh2** | 0.352 | 0.297 | 0.348 | 0.290 | 0.401 | 0.372 | 0.397 | 0.367 | 0.416 | 0.386 | 0.427 | 0.400 | 0.441 | 0.408 | 0.439 | 0.406 | 0.403 | 0.366 | 0.403 | 0.366 |
| **ETTm1** | 0.350 | 0.294 | 0.348 | 0.293 | 0.380 | 0.341 | 0.370 | 0.331 | 0.396 | 0.369 | 0.396 | 0.370 | 0.427 | 0.424 | 0.426 | 0.428 | 0.388 | 0.357 | 0.385 | 0.355 |
| **ETTm2** | 0.263 | 0.172 | 0.260 | 0.169 | 0.297 | 0.226 | 0.300 | 0.226 | 0.346 | 0.287 | 0.344 | 0.288 | 0.394 | 0.375 | 0.396 | 0.377 | 0.325 | 0.265 | 0.325 | 0.265 |
| **ECL** | 0.231 | 0.135 | 0.239 | 0.139 | 0.247 | 0.153 | 0.254 | 0.157 | 0.264 | 0.169 | 0.270 | 0.172 | 0.296 | 0.207 | 0.299 | 0.208 | 0.259 | 0.166 | 0.265 | 0.169 |
| **Exchange** | 0.212 | 0.088 | 0.211 | 0.089 | 0.305 | 0.175 | 0.309 | 0.185 | 0.429 | 0.346 | 0.461 | 0.397 | 0.688 | 0.928 | 0.729 | 0.969 | 0.408 | 0.384 | 0.428 | 0.410 |
| **Solar** | 0.245 | 0.197 | 0.249 | 0.203 | 0.258 | 0.212 | 0.273 | 0.215 | 0.274 | 0.216 | 0.285 | 0.219 | 0.280 | 0.221 | 0.298 | 0.227 | 0.264 | 0.211 | 0.276 | 0.216 |
| **Traffic** | 0.259 | 0.371 | 0.287 | 0.396 | 0.269 | 0.398 | 0.291 | 0.413 | 0.274 | 0.406 | 0.303 | 0.428 | 0.299 | 0.443 | 0.312 | 0.452 | 0.275 | 0.404 | 0.298 | 0.422 |
| **Weather** | 0.199 | 0.151 | 0.205 | 0.156 | 0.243 | 0.196 | 0.245 | 0.198 | 0.248 | 0.250 | 0.286 | 0.251 | 0.337 | 0.324 | 0.337 | 0.325 | 0.257 | 0.230 | 0.268 | 0.232 |
| **PEMS03** | 0.236 | 0.135 | 0.247 | 0.140 | 0.257 | 0.167 | 0.274 | 0.175 | 0.268 | 0.180 | 0.280 | 0.186 | 0.312 | 0.229 | 0.321 | 0.233 | 0.268 | 0.178 | 0.281 | 0.183 |
| **PEMS04** | 0.327 | 0.417 | 0.345 | 0.431 | 0.347 | 0.448 | 0.365 | 0.463 | 0.364 | 0.478 | 0.370 | 0.482 | 0.409 | 0.565 | 0.422 | 0.577 | 0.362 | 0.477 | 0.375 | 0.488 |
| **PEMS07** | 0.216 | 0.104 | 0.227 | 0.113 | 0.232 | 0.127 | 0.246 | 0.135 | 0.237 | 0.137 | 0.248 | 0.143 | 0.258 | 0.177 | 0.289 | 0.190 | 0.235 | 0.136 | 0.252 | 0.145 |
| **PEMS08** | 0.333 | 0.458 | 0.348 | 0.467 | 0.342 | 0.457 | 0.378 | 0.539 | 0.377 | 0.561 | 0.394 | 0.575 | 0.383 | 0.564 | 0.428 | 0.649 | 0.359 | 0.510 | 0.387 | 0.557 |

The four TSFMs are fine-tuned on downstream datasets to strengthen generalization and ensure comparable parameter distributions before and after pruning, following (Zhao et al., 2026). Full results are reported in Table 25- 28.

*Table 25.* Full results of SUNDIAL_LARGE. Input length is 336, horizons $\in \{96, 192, 336, 720\}$.

| Horizons | 96 | | | | 192 | | | | 336 | | | | 720 | | | | avg | | | |
|---|---|---|---|---|---|---|---|---|---|---|---|---|---|---|---|---|---|---|---|---|
| | FT+P | | FT | | FT+P | | FT | | FT+P | | FT | | FT+P | | FT | | FT+P | | FT | |
| Metric | MAE | MSE | MAE | MSE | MAE | MSE | MAE | MSE | MAE | MSE | MAE | MSE | MAE | MSE | MAE | MSE | MAE | MSE | MAE | MSE |
| **ETTh1** | 0.371 | 0.338 | 0.379 | 0.350 | 0.397 | 0.373 | 0.405 | 0.382 | 0.410 | 0.405 | 0.420 | 0.415 | 0.454 | 0.430 | 0.467 | 0.444 | 0.408 | 0.387 | 0.418 | 0.398 |
| **ETTh2** | 0.331 | 0.265 | 0.343 | 0.274 | 0.362 | 0.315 | 0.370 | 0.318 | 0.397 | 0.346 | 0.411 | 0.358 | 0.439 | 0.380 | 0.437 | 0.384 | 0.382 | 0.327 | 0.390 | 0.334 |
| **ETTm1** | 0.326 | 0.269 | 0.337 | 0.284 | 0.356 | 0.312 | 0.354 | 0.308 | 0.382 | 0.340 | 0.399 | 0.351 | 0.403 | 0.393 | 0.407 | 0.393 | 0.367 | 0.329 | 0.374 | 0.334 |
| **ETTm2** | 0.255 | 0.167 | 0.254 | 0.172 | 0.299 | 0.223 | 0.301 | 0.231 | 0.329 | 0.269 | 0.333 | 0.279 | 0.375 | 0.341 | 0.372 | 0.339 | 0.315 | 0.250 | 0.315 | 0.255 |
| **ECL** | 0.224 | 0.126 | 0.224 | 0.128 | 0.239 | 0.137 | 0.245 | 0.141 | 0.272 | 0.169 | 0.273 | 0.171 | 0.304 | 0.210 | 0.307 | 0.212 | 0.260 | 0.161 | 0.262 | 0.163 |
| **Weather** | 0.148 | 0.203 | 0.153 | 0.205 | 0.197 | 0.245 | 0.203 | 0.252 | 0.255 | 0.287 | 0.256 | 0.293 | 0.309 | 0.325 | 0.319 | 0.334 | 0.227 | 0.265 | 0.233 | 0.271 |

*Table 26.* Full results of CHORONS_BASE. Input length is 336, horizons $\in \{96, 192, 336, 720\}$.

| Horizons | 96 | | | | 192 | | | | 336 | | | | 720 | | | | avg | | | |
|---|---|---|---|---|---|---|---|---|---|---|---|---|---|---|---|---|---|---|---|---|
| | FT+P | | FT | | FT+P | | FT | | FT+P | | FT | | FT+P | | FT | | FT+P | | FT | |
| Metric | MAE | MSE | MAE | MSE | MAE | MSE | MAE | MSE | MAE | MSE | MAE | MSE | MAE | MSE | MAE | MSE | MAE | MSE | MAE | MSE |
| ETTh1 | 0.372 | 0.360 | 0.374 | 0.362 | 0.401 | 0.409 | 0.404 | 0.416 | 0.416 | 0.438 | 0.428 | 0.452 | 0.431 | 0.456 | 0.439 | 0.464 | 0.405 | 0.416 | 0.411 | 0.424 |
| ETTh2 | 0.308 | 0.260 | 0.313 | 0.269 | 0.354 | 0.331 | 0.358 | 0.332 | 0.387 | 0.371 | 0.392 | 0.380 | 0.413 | 0.387 | 0.418 | 0.395 | 0.366 | 0.337 | 0.370 | 0.344 |
| ETTm1 | 0.317 | 0.288 | 0.315 | 0.285 | 0.338 | 0.325 | 0.342 | 0.333 | 0.367 | 0.354 | 0.368 | 0.367 | 0.402 | 0.421 | 0.406 | 0.429 | 0.356 | 0.347 | 0.358 | 0.354 |
| ETTm2 | 0.230 | 0.157 | 0.236 | 0.165 | 0.272 | 0.220 | 0.272 | 0.231 | 0.314 | 0.276 | 0.318 | 0.283 | 0.375 | 0.368 | 0.382 | 0.394 | 0.298 | 0.255 | 0.302 | 0.268 |
| ECL | 0.206 | 0.119 | 0.211 | 0.125 | 0.217 | 0.138 | 0.226 | 0.143 | 0.217 | 0.151 | 0.242 | 0.153 | 0.281 | 0.193 | 0.296 | 0.204 | 0.230 | 0.150 | 0.244 | 0.156 |
| Weather | 0.177 | 0.145 | 0.196 | 0.163 | 0.224 | 0.191 | 0.244 | 0.223 | 0.259 | 0.236 | 0.279 | 0.261 | 0.322 | 0.316 | 0.350 | 0.381 | 0.246 | 0.222 | 0.267 | 0.257 |

*Table 27.* Full results of MOIRAI_LARGE. Input length is 336, horizons $\in \{96, 192, 336, 720\}$.

| Horizons | 96 | | | | 192 | | | | 336 | | | | 720 | | | | avg | | | |
|---|---|---|---|---|---|---|---|---|---|---|---|---|---|---|---|---|---|---|---|---|
| | FT+P | | FT | | FT+P | | FT | | FT+P | | FT | | FT+P | | FT | | FT+P | | FT | |
| Metric | MAE | MSE | MAE | MSE | MAE | MSE | MAE | MSE | MAE | MSE | MAE | MSE | MAE | MSE | MAE | MSE | MAE | MSE | MAE | MSE |
| ETTh1 | 0.396 | 0.372 | 0.402 | 0.377 | 0.415 | 0.411 | 0.416 | 0.414 | 0.428 | 0.457 | 0.437 | 0.461 | 0.447 | 0.460 | 0.459 | 0.478 | 0.422 | 0.425 | 0.429 | 0.433 |
| ETTh2 | 0.344 | 0.283 | 0.342 | 0.285 | 0.389 | 0.346 | 0.393 | 0.352 | 0.405 | 0.371 | 0.402 | 0.370 | 0.430 | 0.393 | 0.427 | 0.398 | 0.392 | 0.348 | 0.391 | 0.351 |
| ETTm1 | 0.343 | 0.311 | 0.344 | 0.312 | 0.369 | 0.345 | 0.375 | 0.351 | 0.382 | 0.358 | 0.391 | 0.366 | 0.402 | 0.395 | 0.414 | 0.398 | 0.374 | 0.352 | 0.381 | 0.357 |
| ETTm2 | 0.246 | 0.161 | 0.240 | 0.159 | 0.285 | 0.223 | 0.292 | 0.227 | 0.334 | 0.286 | 0.341 | 0.293 | 0.387 | 0.368 | 0.395 | 0.371 | 0.313 | 0.260 | 0.317 | 0.263 |
| ECL | 0.231 | 0.135 | 0.239 | 0.146 | 0.263 | 0.165 | 0.271 | 0.174 | 0.284 | 0.182 | 0.290 | 0.183 | 0.298 | 0.214 | 0.296 | 0.214 | 0.269 | 0.174 | 0.274 | 0.179 |
| Weather | 0.202 | 0.144 | 0.211 | 0.152 | 0.240 | 0.197 | 0.249 | 0.208 | 0.312 | 0.263 | 0.311 | 0.259 | 0.374 | 0.349 | 0.397 | 0.394 | 0.282 | 0.238 | 0.292 | 0.253 |

*Table 28.* Full results of TIMESTM_200M. Input length is 336, horizons $\in \{96, 192, 336, 720\}$.

| Horizons | 96 | | | | 192 | | | | 336 | | | | 720 | | | | avg | | | |
|---|---|---|---|---|---|---|---|---|---|---|---|---|---|---|---|---|---|---|---|---|
| | FT+P | | FT | | FT+P | | FT | | FT+P | | FT | | FT+P | | FT | | FT+P | | FT | |
| Metric | MAE | MSE | MAE | MSE | MAE | MSE | MAE | MSE | MAE | MSE | MAE | MSE | MAE | MSE | MAE | MSE | MAE | MSE | MAE | MSE |
| ETTh1 | 0.386 | 0.375 | 0.394 | 0.388 | 0.413 | 0.415 | 0.414 | 0.412 | 0.411 | 0.418 | 0.417 | 0.425 | 0.432 | 0.430 | 0.437 | 0.434 | 0.411 | 0.410 | 0.416 | 0.415 |
| ETTh2 | 0.323 | 0.261 | 0.331 | 0.266 | 0.359 | 0.326 | 0.365 | 0.331 | 0.386 | 0.354 | 0.389 | 0.358 | 0.406 | 0.386 | 0.418 | 0.394 | 0.369 | 0.332 | 0.376 | 0.337 |
| ETTm1 | 0.332 | 0.315 | 0.336 | 0.325 | 0.363 | 0.359 | 0.361 | 0.362 | 0.387 | 0.376 | 0.399 | 0.408 | 0.407 | 0.413 | 0.418 | 0.425 | 0.372 | 0.366 | 0.379 | 0.380 |
| ETTm2 | 0.247 | 0.164 | 0.252 | 0.165 | 0.290 | 0.229 | 0.291 | 0.228 | 0.306 | 0.271 | 0.314 | 0.286 | 0.362 | 0.351 | 0.369 | 0.355 | 0.301 | 0.254 | 0.307 | 0.259 |
| ECL | – | – | – | – | – | – | – | – | – | – | – | – | – | – | – | – | – | – | – | – |
| Weather | – | – | – | – | – | – | – | – | – | – | – | – | – | – | – | – | – | – | – | – |

# F. Discussion

**Discussion 1: From hidden states to prediction.**    Building on our finding that model still performs well after pruning most layers with fine-tuning, we further analyze how temporal hidden states evolve across layers. The hidden states from

each layer are fed into Prediction head to generate forecasts, as shown in Fig. 15.

For reference, (Cols. 1 & 4) correspond to the unpruned model, where Prediction head consumes hidden states from the initial state (input embedding with positional encoding only) and final layer. By preserving only the first and last layers (Cols. 3 & 6) without any fine-tuning, the model still delivers competitive performance. In contrast, projecting intermediate layers back to the TS yields outputs that diverge from ground-truth distribution (Cols. 2 & 5).

The representations in layers 2~10 exhibit substantial deviation from the TS domain, whereas the first and last layers primarily function as domain adapters between the TS input and the model's internal representations. Consequently, even when only layers 0 and 11 are retained and all others are pruned, the pruned model still delivers satisfactory performance without any fine-tuning.

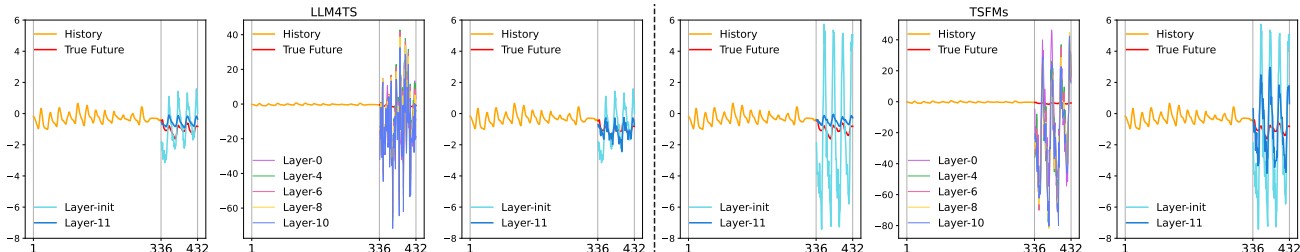

*Figure 15.* Visualization of hidden states projected into the TS domain. Cols. 1~3 show results from LLM4TS models, and Cols. 4~6 present those from TSFMs.

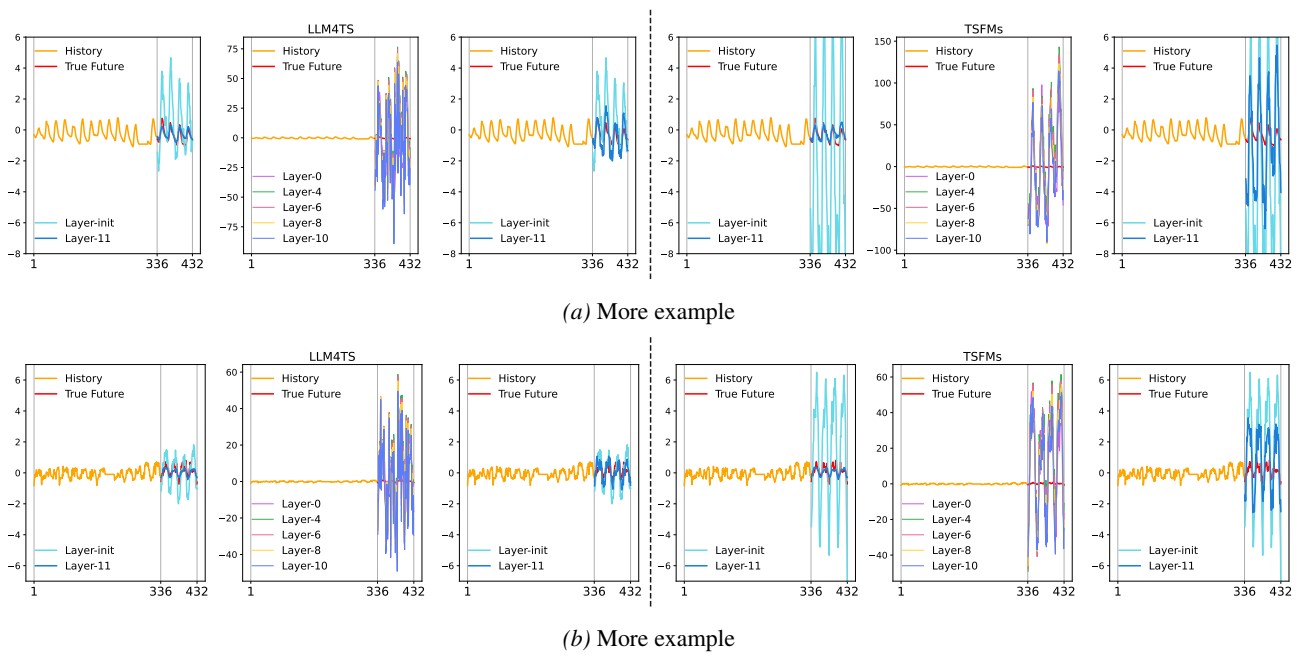

Discussion 2: Attention weights as a window into pattern learning for TS. Attention entropy $\mathcal{H}^l$ measures the attention distribution of each head across tokens (Zhang et al., 2025). Interestingly, we find that $\mathcal{H}^l$ remains stable across layers in Fig. 17, meaning limited change in attention dispersion. Moreover, heads often focus on diagonal positions, which exert dominant influence on prediction in Fig. 18. If very few tokens determine the prediction, it is understandable that a small set of informative token features may suffice without extensive parameters (Kim et al., 2022; Wen et al., 2025).

**Discussion 3: Temporal Structure Shapes Representation Dynamics.** To characterize the temporal structures underlying our analysis, we probe the model at inference time using input sequences with different correlation properties and examine their impact on representation dynamics across depth. As shown in Fig. 19, strongly correlated sequences exhibit smooth

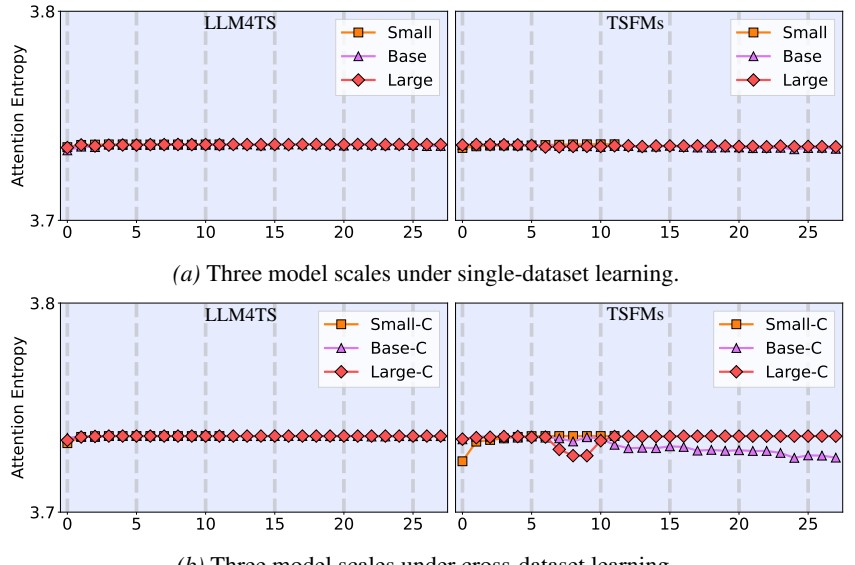

*(a)* Three model scales under single-dataset learning.

*(b)* Three model scales under cross-dataset learning.

*Figure 17.* Attention entropy of all layers across twelve models. All layers exhibit nearly identical entropy values (3.7~3.8), suggesting minimal diversity in attention distribution.

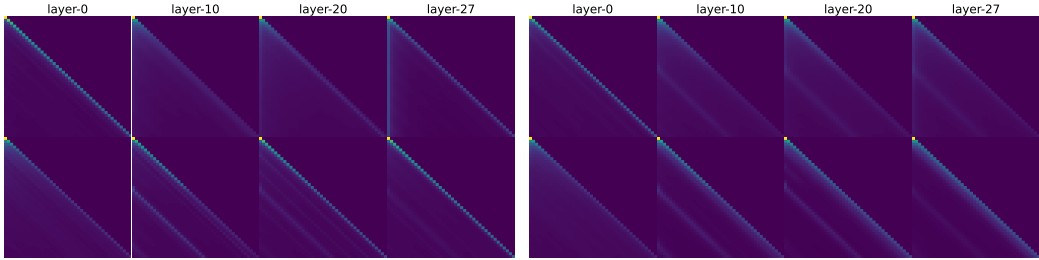

*Figure 18.* Mean attention weights across all heads for four models. The attention is primarily focused on the diagonal, with only a few tokens being critical for prediction. (***Upper-left***) attention weights of LLM4TS-Base, (***Lower-left***) attention weights of TSFMs-Base, (***Upper-right***) attention weights of LLM4TS-Large, (***Lower-right***) attention weights of TSFMs-Large.

temporal evolution with high redundancy across adjacent time steps, whereas weakly correlated but structured sequences display more complex and non-stationary patterns, and white noise lacks meaningful temporal dependency. We then quantify representation dimensionality using $m_{90}$, defined as the number of principal components required to explain 90% of the variance. As shown in Fig. 20, strongly correlated inputs lead to a rapid reduction of $m_{90}$ in shallow layers, indicating early concentration of representations into a low-dimensional subspace. In contrast, weakly correlated but structured sequences maintain higher dimensionality over a larger depth range, while white noise preserves high $m_{90}$ across layers. Taken together, these observations indicate that strong temporal correlation in TS data promotes early representational saturation, which in turn contributes to layer-wise redundancy in deep TS Transformers.

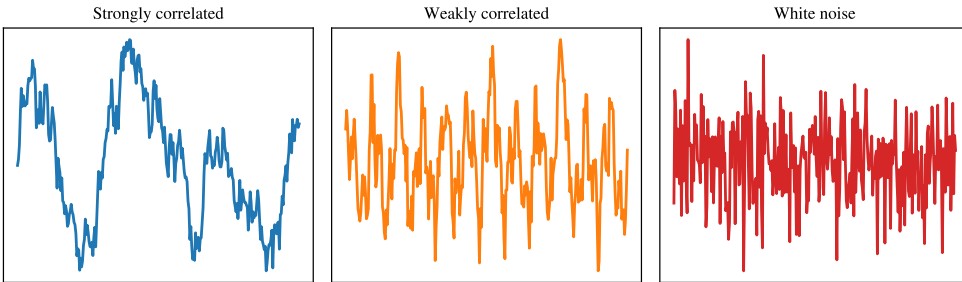

*Figure 19.* Representative time-domain signals with different temporal structures: strongly correlated, weakly correlated but structured, and white noise.

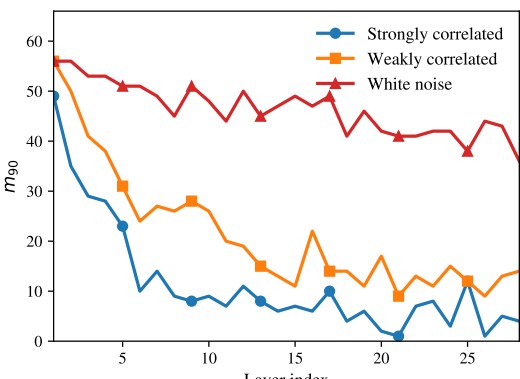

*Figure 20.* Layer-wise evolution of representation dimensionality measured by $m_{90}$, the number of principal components required to explain 90% of the variance, under different temporal structures.

