# OpenReview forum: "Revealing Scaling Paradox in Large-scale Time Series Models: Implications for More Efficient and Accurate Forecasting"
_ICML.cc/2026/Conference — ICML 2026 regular_

### Official Review · Reviewer_gEyF · 2026-03-03

**Soundness:** 3
**Presentation:** 2
**Significance:** 3
**Originality:** 3
**Overall Recommendation:** 4
**Confidence:** 4

**Summary:**

This paper investigates the "scaling paradox" in time series (TS) forecasting models, challenging the prevailing assumption that scaling up model capacity and data volume consistently improves performance. The authors conduct a comprehensive empirical study on two dominant paradigms: fine-tuning text-based Large Language Models for TS (LLM4TS) and training Time Series Foundation Models (TSFMs) from scratch. Through extensive experiments across multiple model scales (100M to 1.7B parameters) and massive datasets (up to 6B observations), the paper verifies the pervasive nature of this scaling paradox. Furthermore, by analyzing internal representations, the authors diagnose the root cause as "few-layer dominance"—where only a small subset of layers contributes meaningfully, while the rest are redundant or even detrimental. Consequently, the authors propose a practical method to automatically identify and retain only these dominant layers, demonstrating significant improvements in both accuracy (up to 12%) and inference speed (2.7×) while retaining only a fraction of the parameters.

**Compliance With Llm Reviewing Policy:**

Affirmed.

**Final Justification:**

My concerns have been addressed, particularly regarding the open-source issues and which specific layers have been retained. I keep my positive score.

**Key Questions For Authors:**

Result Inconsistencies: Could you please explain the discrepancies in the reported results? Specifically, why do the LLM4TS "Base" results differ so significantly between Table 13 and Table 14 (ratio 1.0), and why do the results for TSFMs tiny/large in Table 13 not match Table 15 under the same settings(ratio 1.0)?

LLM4TS Baselines vs. Proposed Method: In Table 6, are your proposed method and the baseline methods (which also truncate layers) evaluated using the exact same number of retained layers?

Important Layers vs. Naive Truncation: Could you explicitly list which specific layers are typically retained by your method? If the selected layers are mostly the early/last layers (as is common in LLMs and suggested by your appendix), how does your proposed selection method quantitatively compare to a "naive truncation" baseline (e.g., simply keeping the first few layers)?

Screening vs. Training Data: Could you clarify the dataset splitting strategy? Specifically, which datasets are used to calculate layer importance scores, and which datasets are used for subsequent training?

Open Source and Reproducibility: A major contribution of this work is establishing a fair and comprehensive evaluation environment. Do you plan to open-source the processed datasets, as well as the complete code for pruning, training, and fine-tuning upon publication?

**Limitations:**

The paper does not discuss shortcomings. It is recommended to compare performance with deep pruning methods for LLMs, such as LLMStreamline. The paper states that it will not have adverse effects on society.

**Strengths And Weaknesses:**

Strengths：
Comprehensive and Rigorous Experimental Setup: I really appreciate the experimental design of this paper. The evaluation is conducted across multiple dimensions, including data volume, data distribution, model categories, and model sizes. This provides a very comprehensive and fair evaluation environment, which is a significant contribution to the time series forecasting community.

Valuable Exploration of Scaling Laws: The systematic exploration of scaling laws (and the scaling paradox) in the time series domain is timely and highly valuable. The empirical findings provide important insights for future research in large time-series models.

Insightful Representation Analysis: The analysis of layer representations and the identification of the "few-layer dominance" phenomenon are well-executed. The layer similarity visualization in the appendix is particularly interesting and supports the claims well.

Practical and Efficient Solution: The proposed method of pruning/fine-tuning requires fewer resources, accelerates inference, and preserves (or even improves) performance compared to full fine-tuning.

Weaknesses see Key Questions。

---

> ### Author Rebuttal · Authors · 2026-03-31
>
> We thank the reviewer for the thoughtful and positive assessment of our paper. We especially appreciate the recognition of our comprehensive experimental setup, the value of our scaling-law exploration in time series forecasting, the insightful layer-representation analysis, and the practical efficiency of our pruning/fine-tuning approach. We respond to the weaknesses and questions point by point below.
>
> ---
> [Q1]
>
> - We thank the reviewer for the careful check. The results in Tab 13 are the correct version, and the corresponding figure in the main text was also plotted using these data. The corresponding entries in Tables 14 and 15 contain formatting misalignment. We will correct them in the revision.
>
> ---
> [Q2]
>
> - It is not the same quantity. For the baseline models, we follow their original settings without modification, and their architectures differ substantially. In contrast, our method applies a more aggressive pruning strategy under a unified framework. Despite this, it achieves better performance in most cases, demonstrating a more favorable trade-off between model capacity and efficiency.
>
> ---
> [Q3]
>
> - The retained layers are primarily determined by the score, and the early/last layers are only part of them, as shown in Fig. 14, Average layer-wise importance. In fact, our layer retention is principled rather than based on arbitrary selection. Moreover, the number of retained layers is smaller than that of the baseline, which is also reflected in the practical SP: Average Speedup Ratio.
>
> - The specific retained layers under different settings are listed in the tab 1.
>
> Tab1:Specific layers typically retained by our method.
>
> | Setting            | Retained Layers            |
> | ------------------ | -------------------------- |
> | (a) LLM4TS-Small   | 2,0,1,11,10                |
> | (b) LLM4TS-Base    | 2,27,0,1,26,25,24,23,21,22 |
> | (c) LLM4TS-Large   | 1,27,2,3,0,23,19,4,5,13,6  |
> | (d) TSFMs-Small    | 0,7,8,5,9                  |
> | (e) TSFMs-Base     | 0,3,2,8,27,26,25,11,24     |
> | (f) TSFMs-Large    | 0,27,5,3,1,15,2,4,12,9,11  |
> | (g) LLM4TS-Small-C | 2,1,0,11,5                 |
> | (h) LLM4TS-Base-C  | 2,1,0,27,23,22,3,14,9,6,8  |
> | (i) LLM4TS-Large-C | 2,1,0,27,25,18,6,8,11      |
> | (j) TSFMs-Small-C  | 0,3,4,6                    |
> | (k) TSFMs-Base-C   | 0,4,3,26,2,8,25,24,20      |
> | (l) TSFMs-Large-C  | 8,0,7,26,6,10,25,10,9,24   |
>
> - We also observe that even with the same backbone, the layers deemed important differ between single-dataset and cross-dataset training, with non-negligible discrepancies. This suggests that static truncation or retaining a fixed set of layers is insufficient.
>
> ---
> [Q4]
>
> - We follow the same dataset split strategy as prior work, namely a 7:1:2 split for training, validation, and testing. The layer importance scores are computed only on the validation set, while the test set is used exclusively for final evaluation. For all subsequent training stages, as well as the initial fine-tuning of TSFMs, only the training set is used.
>
> - Due to the strong heterogeneity across time-series domains(Fig. 11  Statistical characteristics of datasets, including seasonality, trend, stationarity, transition, shifting, correlation, and nonGaussianity.), computing layer importance on a single dataset can be biased and unreliable. To improve robustness, we estimate importance on a shared validation set covering diverse data characteristics, and use the selected layers for subsequent training to promote better cross-domain generalization.
>
> ---
> [Q5]
>
> - Yes. We plan to release all processed datasets and the full codebase for pruning, training, and fine-tuning upon publication. In fact, we are already organizing these materials for public release, including the data preprocessing pipeline, experiment scripts, and implementation details needed to reproduce our results. Our goal is to make the benchmark and the proposed method as transparent and reusable as possible for the community.
>
> ---
> [Limitation]
>
> - The main limitation of our study is that the analysis is conducted within currently mainstream model architectures and training paradigms, whereas the field is evolving quickly and new approaches continue to emerge.
>
> - We note, however, that methods such as LLMStreamline are developed for generic LLM pruning, whereas our study focuses on time series forecasting foundation models and aims primarily to investigate the scaling paradox and the few-layer dominance phenomenon, rather than to propose a dedicated pruning method. Therefore, such a comparison is not entirely aligned with the main purpose of this work. Nevertheless, we believe that, under the theme of exploiting the advantages of a small subset of layers, conducting a more fine-grained comparison with different methods would be meaningful.
>
> ---
> If there are any further questions, please feel free to contact us, as your feedback is important for improving the quality of our manuscript.

---

> > ### Author Rebuttal · Reviewer_gEyF · 2026-04-02
> >
> > Thanks for your response. My concerns have been addressed, particularly regarding the open-source issues and which specific layers have been retained. I keep my positive score.

---

> > > ### Author Response · Authors · 2026-04-06
> > >
> > > We sincerely thank the reviewer for the thorough and constructive feedback, as well as for the recognition of our work. We appreciate the time and effort invested in the review.

---

### Official Review · Reviewer_7GAa · 2026-03-06

**Soundness:** 3
**Presentation:** 2
**Significance:** 3
**Originality:** 3
**Overall Recommendation:** 4
**Confidence:** 4

**Summary:**

The paper studies the scaling behavior of mainstream large TS models, revealing a scaling paradox phenomenon, which is a well-known and important problem in this field. By locating the root cause - the failure of most hidden layers/parameter redundancies, the author proposed a trivial pruning strategy for large TS models.

**Compliance With Llm Reviewing Policy:**

Affirmed.

**Final Justification:**

The overall quality of the research is relatively high, although I have some initial concerns about the originality and the presentation. That said, some of the main conclusions of the research, in my opinion, have already been discussed by existing works. I think the authors' responses have addressed most of my concerns, so I decided to raise my recommendation to a weak accept.

**Key Questions For Authors:**

- Q1: When pruning some less informative layers, how to ensure the matches of input-output dimension/neurons of different hidden layers?

- Q2: It shows that by dropping a few layers of large models, many forecasting results suffer from considerable loss (e.g., xx), which raises concerns about the practical use. Is this a continuous space of pruning layers?  In other words, is there any performance variation trend across different pruning rates, by which one can control the performance and efficiency trade-off?

- Q3: If deep layers are redundant, what are the advantages of these large models compared with small models?

**Limitations:**

While the future work is mentioned, the conclusion, limitations are not discussed.

**Strengths And Weaknesses:**

Strengths:
- The paper is well written and organized.
- The identified "paradox“ seems to be valid, and the proposed pruning is useful for improving both accuracy and efficiency.
- The experiments and analyses are comprehensive.
- The implementation details are provided.

Weaknesses:
- The problem is not new in this field, as the author conducted their explorations mainly on commonly used eight datasets (ETTh, Weather ...), which were already known to be non-scalable datasets and just an important data resource for forecasting evaluation. The author uses a large portion of the text to emphasize the so-called "scaling paradox". However, the model redundancy when applying DL to time series forecasting is already revealed by existing research (e.g., [1] identifies the structure redundancy through compression; [2] uses orthogonal transform to ensure high-rankness, inherently reducing the redundancy); that is to say, many of the explorations in this manuscript are somehow repetitive (e.g., the RQ1 and RQ2) and cannot bring much new information to the community.

- Due to the above reason, in my opinion, the most valuable contribution should lie in the technique that proposes to resolve or make use of the analysed root cause. However, the proposed pruning method is somewhat brute yet not deep enough, and the experiment exploration shares the same problem, i.e., not thorough, which lowers the overall quality of this study. For example, as research related to the **scaling capacity** of large time series models, apart from leveraging the paradox to reduce the volume of the model, what benefits the community a lot may be the potential solution or suggestions to unlock the revealed bottleneck. What is the effect of shrinking the parameters by reducing the NN's width instead of shortening the length?

- I think the manuscript is overdecorated. I suggest to use less colors (e.g., light yellow) that are too transparent and make it hard to read.
There are some typos: The first letter followed by the abbreviation should be in capital; "cross-dataset" in line 169 should be "Cross".

- Some arguments are arbitrary, e.g., in Takeaway 3, "In out-of-domain testing, larger models show no advantage.".

- The reference of the paper “palm: Scaling language ...” is not right.

[1] Lu, Yihang, et al. "Timecapsule: Solving the jigsaw puzzle of long-term time series forecasting with compressed predictive representations." Proceedings of the 31st ACM SIGKDD Conference on Knowledge Discovery and Data Mining V. 2. 2025.

[2] Yue, Wenzhen, et al. "Olinear: A linear model for time series forecasting in orthogonally transformed domain." arXiv preprint arXiv:2505.08550 (2025).

---

> ### Author Rebuttal · Authors · 2026-03-31
>
> We thank the reviewer for the assessment of our paper. We respond to the weaknesses and questions point by point below.
>
> ---
> [W1/W2]
>
> - Regarding the data issue, the eight commonly used datasets are only one part of our study. Our conclusions are further extended to a broader evaluation on 40+ datasets; see **our response to Reviewer ZvJM [W1]** for details.
>
> - More importantly, the scaling paradox studied in this paper is not the same as the already known observation of model redundancy. Few-layer dominance is our explanation of this phenomenon. Accordingly, RQ1 and RQ2 are not redundant: RQ1 examines whether the phenomenon consistently holds across different architectures and model scales, while RQ2 tests whether it disappears as more in-distribution data become available. Rather than re-showing that models are compressible, these analyses aim to clarify whether the scaling paradox is a general pattern, a data-limited effect, or an artifact of specific training settings. We believe this scaling-oriented analysis provides useful insights into the design and scaling of time series foundation models.
>
> - Compared with earlier Chronos versions, Amazon’s recently released Chronos-2 achieves highly competitive, and in some cases state-of-the-art, performance despite a substantial reduction in parameter count. This, to some extent, further reinforces the central concern highlighted in our work.
>
> - In general, reducing the depth of a neural network can more directly lower computational cost and is also more closely related to the few-layer dominance phenomenon. By contrast, reducing the network’s width often does not lead to direct speedup if done in a non-structured manner; if performed through structured removal, it poses a greater challenge for preserving the capabilities of pretrained models. Even so, we still think this a promising direction.
>
> ---
> [W3-W5]
>
> - We thank the reviewer for these helpful details. The current reference was taken from the official GitHub repository, which appears to omit some details, and we will correct them in the revision. More precisely, “show no expected advantage” is a more appropriate.
>
> ---
> [Q1-Q3]
>
> - Our pruning is performed at the whole-layer level, and the retained layers remain unchanged in their hidden size and interface. This greatly simplifies our approach, as it avoids dimension mismatch issues and eliminates the need for additional adaptation modules or retraining.
>
> - Yes. The pruning space is continuous, and as shown in Fig. 8, different pruning rates lead to different accuracy–efficiency trade-offs.
> In other words, performance does not collapse only at a single pruning point; rather, it varies progressively as more layers are removed. This provides a controllable spectrum between efficiency and accuracy, allowing users to select a pruning rate according to practical deployment constraints. At the same time, our results also suggest that overly aggressive pruning can cause noticeable degradation, which highlights the need to choose the pruning rate carefully depending on the target scenario.
>
> - Its advantage is more likely to emerge in out-of-domain or more open-ended scenarios, where training and deploying a separate highly specialized small model for each specific setting is often impractical and uneconomical. Large models provide stronger capacity for generalization and adaptability under distribution shifts, even if some layers may appear redundant in specific in-domain settings. At the same time, how to better utilize this capacity when scaling models, or how to train models that achieve strong generalization while maintaining a manageable size, remains an important direction for the community.
>
> ---
> [Limitation]
>
> - The conclusion and future work have already been discussed in the manuscript. The main limitation is that our analysis is still conducted within mainstream model architectures and training paradigms, while the field is evolving rapidly and new approaches continue to emerge.
>
> ---
> If there are any further questions, please feel free to contact us, as your feedback is important for improving the quality of our manuscript.

---

> > ### Author Rebuttal · Reviewer_7GAa · 2026-04-02
> >
> > I appreciate the authors' clarifications and will raise my score.

---

> > > ### Author Response · Authors · 2026-04-06
> > >
> > > We thank the reviewer for the insightful feedback and kind acknowledgment of our contributions. We greatly appreciate the time and attention dedicated to reviewing our paper.

---

### Official Review · Reviewer_d3eW · 2026-03-13

**Soundness:** 3
**Presentation:** 4
**Significance:** 3
**Originality:** 3
**Overall Recommendation:** 5
**Confidence:** 4

**Summary:**

This paper investigates the scaling behavior of large time series forecasting models and reports a phenomenon termed the Scaling Paradox, where increasing model size does not consistently improve forecasting performance. The authors evaluate LLM4TS and TSFMs across model sizes ranging from 100M to 1.7B parameters, under both Single-dataset Learning and Cross-dataset Learning settings, and across In-domain and Out-of-domain evaluations.
Through empirical analysis of inter-layer and intra-layer representations, the authors observe a phenomenon named Few-layer Dominance, suggesting that only a small subset of layers significantly contributes to predictive performance. Based on this observation, they propose a Critical-layer Identification Method to prune redundant layers, reporting comparable or improved accuracy with substantial inference speedups.
This paper challenges the assumption that scaling laws observed in NLP directly transfer to time series forecasting.

**Compliance With Llm Reviewing Policy:**

Affirmed.

**Final Justification:**

I acknowledge the contributions of this work. After a careful review, I decide to put the score to be 5, i.e., accept.  I hope the authors will incorporate my suggestions into the final version to further improve the paper's quality.

**Key Questions For Authors:**

1. What’s the essential difference between Few-layer Dominance and existing studies on layer redundancy?
2. Is fine-tuning after retaining the critical layer a necessary step? Did the fine-tuning have a significant impact on the experimental results?
3. Can it be proven that the redundancy layer always exists regardless of the size of the dataset?

**Limitations:**

The discussion on limitations is insufficient. The paper only covers channel-independent forecasting and does not cover multi-channel forecasting.

**Strengths And Weaknesses:**

Strengths
1. This paper demonstrates that in the field of TSF using large-scale models, larger models do not necessarily lead to better prediction results, and this conclusion has practical significance.
2. This paper evaluates multiple model families, scales, and learning strategies. The experimental scope is ambitious and attempts to provide a comprehensive view of scaling behavior in time series forecasting.
3. The method proposed in this paper for preserving dominant layers in the model effectively improves inference efficiency and has practical significance.

Weaknesses
1. The proposed Critical-layer Identification Method is the most important component of this paper. However, its level of novelty appears limited, as it essentially performs importance ranking followed by pruning. The authors do not clearly articulate how this method is fundamentally different from existing layer importance evaluation approaches.

2. The paper argues that the Scaling Paradox is caused by layer redundancy, based on the observation that pruning non-critical layers does not degrade model performance. However, I find this causal claim insufficiently supported. For example, if substantially larger-scale training data were available, would the currently observed redundant layers still exhibit near-identity mappings?The experiments in the paper show that the Scaling Paradox phenomenon is alleviated as the data volume increases. However, due to limitations in dataset size, experiments with significantly larger-scale data cannot be conducted. As a result, it remains unclear whether the emergence of redundant layers is driven by limitations in data scale rather than model scale. At present, redundancy appears to be an outcome of the observed phenomenon rather than its underlying cause.

3. The contribution of this paper leans more toward engineering optimization. It improves inference efficiency without sacrificing performance, which is practically valuable. However, I do not see a clear theoretical breakthrough regarding scaling laws.

---

> ### Author Rebuttal · Authors · 2026-03-31
>
> We thank the reviewer for the careful reading and constructive feedback. We appreciate the recognition of the practical significance of our empirical finding, the broad experimental coverage across models and settings, and the efficiency gains brought by preserving dominant layers.
>
>
> We respond to the weaknesses and questions point by point below.
>
> ---
>
> [W1/Q1]
>
> - We thank the reviewer for this comment. We would like to clarify that the core contribution of this work is not to propose a standalone pruning algorithm, but rather to reveal the scaling paradox in time series forecasting and the underlying phenomenon of few-layer dominance. The Critical-layer Identification Method serves only as an intuitive validation tool derived from this finding.
>
> - More importantly, we hope this work conveys a broader message to the community: in TS forecasting, the performance bottleneck may not necessarily stem from insufficient model scale, but rather from the underutilization of effective layers and redundancy in deep representations. Accordingly, rather than continuing to scale models blindly, it may be more worthwhile to re-examine how depth is actually utilized and how architectures should be designed to make better use of it.
>
> - Moreover, our Few-layer Dominance is a peculiar phenomenon identified in TS models under the scaling paradox. In text LLMs or VLMs, directly removing a majority of layers while incurring almost no performance loss is rare; maintaining performance there usually requires more sophisticated token pruning, adaptive routing, or structure-aware compression schemes[1-3].
>
> ref:
>
> [1]Skipgpt: Dynamic layer pruning reinvented with token awareness and module decoupling. arXiv preprint arXiv:2506.04179. ICML 2025
>
> [2]Llava-prumerge: Adaptive token reduction for efficient large multimodal models. CVPR 2025
>
> [3]Atp-llava: Adaptive token pruning for large vision language models. CVPR 2025
>
> ---
>
> [W2/Q3]
>
> - Our conclusion is not based solely on the observation that pruning non-critical layers does not hurt performance. Rather, it is supported by three consistent pieces of evidence:
>
> (1) as model depth increases, many layers exhibit highly similar representations and near-identity-like behavior;
>
> (2) this pattern repeatedly appears across different architectures, model scales, and learning strategies;
>
> (3) retaining only the few layers identified as critical is often sufficient to preserve, or even improve, forecasting performance. Taken together, these results indicate substantial under-utilization of depth in current TS models, rather than an accidental compression effect.
>
> - Regarding the reviewer’s question of whether the observed redundancy is simply due to insufficient data scale, this is a valuable point. However, our current evidence suggests that the phenomenon is not confined to small-data settings. Beyond the controlled analysis in Sec. 3.2–3.3 on approximately 33M observations, Sec. 3.4 further extends the study to 41 datasets with up to 6B observations. Moreover, in our experiments, the TSFMs we evaluate were themselves pretrained on 100B–1000B observations (**Please also see our response to Reviewer ZvJM [W1]**.). We continue to observe similar patterns under these larger-scale settings. Therefore, the current evidence supports that increasing data scale does not fundamentally eliminate the redundancy reflected by few-layer dominance.
>
> ---
>
> [W3]
>
> - Our work suggests that, in the time series domain, performance bottlenecks may not necessarily arise from insufficient model scale, but rather from the fact that model depth has not been truly utilized effectively. This finding may provide useful insights for the future architectural design of time series foundation models, the development of training strategies, and the study of scaling behavior.
>
> - More specifically, to the best of our knowledge, Amazon’s recently released Chronos-2 achieves highly competitive, and even state-of-the-art, performance despite a substantial reduction in parameter count. This, to some extent, also reinforces the central concern highlighted in our work.
>
> ---
>
> [Q2]
>
> - Yes, FT is necessary and materially affects the results, as shown in Tab.9 of our paper.
>
> ---
>
> [Limitation]
>
> - It mainly arises because multi-channel modeling is not feasible under cross-dataset mixed training, and prior study has also shown that inter-channel dependencies are weak in many mainstream datasets[1].
>
> ref:
>
> [1]A closer look at transformers for time series forecasting: Understanding why they work and where they struggle. ICML 2025
>
>
>
> ---
> If there are any further questions, please feel free to contact us, as your feedback is important for improving the quality of our manuscript.

---

> > ### Author Rebuttal · Reviewer_d3eW · 2026-04-04
> >
> > The authors explained the significance of their work very well and resolved my questions, so I decided to raise my score. I look forward to seeing the corresponding revisions incorporated into the final version of the paper.

---

> > > ### Author Response · Authors · 2026-04-06
> > >
> > > We are grateful to the reviewer for the detailed comments and positive evaluation of our work. Thank you for the time and careful consideration given to our submission.

---

### Official Review · Reviewer_ZvJM · 2026-03-21

**Soundness:** 3
**Presentation:** 3
**Significance:** 3
**Originality:** 3
**Overall Recommendation:** 5
**Confidence:** 4

**Summary:**

The paper analyzes two type of foundation models for time series, fine tuned LLMs and models trained from scratch. Authors show that scaling laws don't always hold for time series and performance can degrade with larger models. This is partly explained by the phenomenon that only a subset of layers contribute to learning/prediction and the rest are redundant or underutilized. Authors then derive a procedure to remove a large subset of layers during prediction leading faster inference and improvement in accuracy.

**Compliance With Llm Reviewing Policy:**

Affirmed.

**Key Questions For Authors:**

Do you have results on newer TS methods such as Chronos 2?

I couldn't find what data is used to compute the quantities in Section 4. Is it all training data or a subset, and how sensitive/stable are the quantities relative to data size?

**Limitations:**

Yes

**Strengths And Weaknesses:**

Strengths: Paper is very well written and easy to follow. Evaluation is through with multiple model types, sizes and datasets. The layer contribution analysis is quite interesting and leads to a robust procedure to identify redundant layers that can be removed during inference. When applied to a diverse set of models the pruning procedure leads to consistently strong improvements in performance and inference time.

Weaknesses: Some of the conclusions are overly strong and should be tempered. The full TS dataset that authors use is only 33M records, which is tiny compared to typical data sizes that foundation models such as LLMs are trained on. I think the only conclusion that can be drawn from this is that at this magnitude of data there is no scaling behavior (there wouldn't be in LLMs either), this should be emphasized. Moreover, models that authors apply their pruning method to are dated and it would be good to see if it still works on newer stronger methods like Chronos 2.

---

> ### Author Rebuttal · Authors · 2026-03-31
>
> We sincerely thank the reviewer for the positive assessment of our work. We are encouraged that the reviewer finds the paper well written and easy to follow, and appreciates the thorough evaluation, the layer contribution analysis, and the effectiveness of our pruning method across diverse models in improving both forecasting performance and inference efficiency.
>
> Below, we respond to each concern and question point by point.
>
> ---
> [W1]The response to the concern that some of the conclusions are overly strong.
>
> - We thank the reviewer for raising this point and appreciate the opportunity to clarify it. In particular, we believe that the statement “the full TS dataset is only 33M records” may not fully reflect the overall scope of our experiments.
>
> - Specifically, in Secs. 3.2 and 3.3, we intentionally start with a controlled setting involving approximately 33M observations from 9 distributions (see main paper, line 148, right column), as this part is intended to serve as an initial analysis. In Sec. 3.4, we further broaden the study to 41 datasets comprising up to 6B observations (see main paper, lines 246–256, left column). As also noted by **Reviewer gEyF in the summary**, our experiments indeed involve “massive datasets (up to 6B observations).”
>
> - In addition, the phenomenon is also supported by our experiments on strong TSFMs that were pretrained on extremely large-scale data. As shown in Table 12 of the main paper (or the table below), the TSFMs in our transfer study were pretrained on 100B, 200B, 230B, and up to 1,000B observations. Despite these substantial pretraining scales, there remains clear room for improvement, and our critical-layer identification and pruning method consistently improves performance.
>
> - Therefore, our conclusion is not based solely on a small-data regime, but is supported by evidence from controlled 33M-scale experiments, broader evaluations up to 6B observations, and strong TSFMs pretrained on 100B–1,000B scale data.
>
> Tab1: Pre-training scale of TSFM models.
> |Type|Model|Backbone|Pre-training Scale|Model Size|
> |-|-|-|-|-|
> |TSFMs|Sundial|Layers:24|1,000B|450M|
> |TSFMs|Chronos|Layers:24|200B|200M|
> |TSFMs|Moirai|Layers:24|230B|300M|
> |TSFMs|TimesFM|Layers:50|100B|500M|
>
> ---
> [W2/Q1]
>
> - We thank the reviewer for this valuable suggestion. We would like to clarify that the pruning method is not intended to be the central standalone contribution of the paper. Rather, it is a simple and intuitive validation tool built upon our main finding, namely that the scaling paradox in TS forecasting is closely related to few-layer dominance.
>
> - We additionally evaluated our method on Chronos-2. The results show that, under four forecasting horizons (96, 192,336 and 720), our method continues to achieve favorable performance, demonstrating that the effectiveness of the layer-selection strategy is not limited to earlier baselines.
>
> Tab2:Results on Chronos-2 under four horizons
> |Metric|Pruning+FT-MAE|Pruning+FT-MSE|FT-MAE|FT-MSE|
> |-|-|-|-|-|
> |ETTh1|0.395|0.386|0.403|0.394|
> |ETTh2|0.368|0.339|0.379|0.352|
> |ETTm1|0.344|0.330|0.341|0.334|
> |ETTm2|0.303|0.257|0.311|0.270|
> |ECL|0.250|0.177|0.252|0.174|
> |Weather|0.224|0.237|0.240|0.232|
>
> ---
> [Q2]
>
> - As noted in Appendix E, we reserve 10% of the training data as a validation set, and the quantities in Section 4 (e.g., those used for Fig. 6 and Fig. 7) are computed on this validation set rather than the test set. This design ensures that the test set is used strictly for final evaluation only. The validation split ratio and partition protocol are also consistent with prior work. In practice, the reported quantities are obtained by averaging over all samples in the validation set, which helps ensure their stability and reduces sensitivity to individual batches or samples.
>
> ---
> If there are any further questions, please feel free to contact us, as your feedback is important for improving the quality of our manuscript.

---

> > ### Author Rebuttal · Reviewer_ZvJM · 2026-04-05
> >
> > Thanks for the response, I will keep my score.

---

> > > ### Author Response · Authors · 2026-04-06
> > >
> > > We thank the reviewer for the detailed feedback and for recognizing the value of our work. We also appreciate the time and effort devoted to evaluating our paper.

---

### Decision · Program_Chairs · 2026-04-30

**Decision:**

Accept (regular)

**Comment:**

This paper studies scaling behavior in large time-series forecasting models and argues that simply increasing model size does not reliably improve forecasting performance. It identifies a recurring few-layer dominance phenomenon, where only a subset of layers appears to contribute meaningfully, and shows that retaining these layers can improve both efficiency and, in many cases, accuracy. Reviewers agreed that this is a timely and practically important topic, and they viewed the empirical study as broad, careful, and informative across model families, scales, and datasets.

The main strengths highlighted by reviewers were the systematic investigation of scaling behavior in time-series forecasting, the representation-level analysis supporting the few-layer dominance observation, and the practical value of the resulting pruning-based procedure. Reviewers also found the paper well written overall and appreciated the combination of empirical insight and deployment relevance. The main concerns were that some claims should be stated more carefully, that the pruning component itself is not the primary novelty, and that the paper should more clearly position its conclusions relative to data scale, prior work on redundancy, and newer models.

The rebuttal addressed these concerns well by clarifying the scope of the claims, adding evidence on stronger recent models, and explaining the broader data settings used in the study. On balance, the paper makes a strong and useful contribution by identifying and carefully documenting an important empirical phenomenon in large-scale time-series forecasting, while also providing a simple but effective way to exploit it. I recommend acceptance.